# Multi-epitope vaccine design using *in silico* analysis of glycoprotein and nucleocapsid of NIPAH virus

**Anoop Kumar** [1]*, **Gauri Misra**[1], **Sreelekshmy Mohandas**[2], **Pragya D. Yadav**[2]*

**1** Molecular Diagnostic Laboratory, National Institute of Biologicals, Noida, Uttar Pradesh, India, **2** Maximum Containment Laboratory, ICMR-National Institute of Virology, Microbial Containment Complex, Pashan, Pune, India

* akmeena87@gmail.com (AK); hellopragya22@gmail.com (PDY)

**Data Availability Statement:** All relevant data are within the manuscript and its Supporting Information file.

**Funding:** The author(s) received no specific funding for this work.

## Abstract

According to the 2018 WHO R&D Blueprint, Nipah virus (NiV) is a priority disease, and the development of a vaccine against NiV is strongly encouraged. According to criteria used to categorize zoonotic diseases, NiV is a stage III disease that can spread to people and cause unpredictable outbreaks. Since 2001, the NiV virus has caused annual outbreaks in Bangladesh, while in India it has caused occasional outbreaks. According to estimates, the mortality rate for infected individuals ranges from 70 to 91%. Using immunoinformatic approaches to anticipate the epitopes of the MHC-I, MHC-II, and B-cells, they were predicted using the NiV glycoprotein and nucleocapsid protein. The selected epitopes were used to develop a multi-epitope vaccine construct connected with linkers and adjuvants in order to improve immune responses to the vaccine construct. The 3D structure of the engineered vaccine was anticipated, optimized, and confirmed using a variety of computer simulation techniques so that its stability could be assessed. According to the immunological simulation tests, it was found that the vaccination elicits a targeted immune response against the NiV. Docking with TLR-3, 7, and 8 revealed that vaccine candidates had high binding affinities and low binding energies. Finally, molecular dynamic analysis confirms the stability of the new vaccine. Codon optimization and *in silico* cloning showed that the proposed vaccine was expressed to a high degree in *Escherichia coli*. The study will help in identifying a potential epitope for a vaccine candidate against NiV. The developed multi-epitope vaccine construct has a lot of potential, but they still need to be verified by *in vitro* & *in vivo* studies.

## 1. Introduction

Emerging/remerging infectious viral diseases that have posed a hazard to humanity throughout history [1]. The ongoing COVID-19 outbreak has had a catastrophic impact on both the global economy and health [2]. Numerous additional viruses, in addition to COVID-19, share this mode of transmission [3,4]. In view of this pandemic, the international community must develop countermeasures in advance of virus outbreaks in order to be ready to combat them.

**Competing interests:** The authors have declared that no competing interests exist.

The Hendra virus (HeV) and Nipah virus (NiV), which had previously been responsible for severe and frequently fatal epidemics in both humans and animals, were discovered in Australia and Malaysia, respectively, in the 1990s. HeV is only present in Australia, but recent NiV outbreaks were initially discovered in Bangladesh, India, and the Philippines. According to several studies the mortality rate for the highly fatal zoonotic virus known as the NiV ranges from 40 to 91% [5–8]. Based on the WHO R&D Blueprint list of priority diseases, it is recommended that the development of the NiV vaccine be expedited. According to stage III zoonotic disease classification standards [9,10], NiV infection can transmit to people and result in isolated epidemics.

The first-ever outbreak in southern India occurred in May 2018 in the state of Kerala, with a total of 19 NiV cases, 17 of which were fatal. *Pteropus giganteus* bats from the outbrake regions of the Kozhikode, Kerala, showed positivity for NiV. Since some NiV strains may spread from person to person and have a high mutation rate, there is an increased chance that NiV will spread globally and cause a pandemic [11]. There is presently no vaccine or licensed therapy for the treatment of NiV. While these outbreaks often tend to be modest, we cannot rule out the possibility that the virus will evolve and/or spread. In addition to being severe, these outbreaks also have a negative influence on society. Infection could spread quickly and humanity could experience its most devastating pandemic if a NiV strain evolved to be human-adapted and infected communities in Southeast Asia, where pigs are a major export product and there are high densities of both humans and pigs [9,12,13].

NiV's genome is a non-segmented single-stranded RNA that is around 18.2 kilobases in size and encodes six structural proteins: nucleocapsid (N), glycoprotein (G), matrix protein (M), fusion protein (F), phosphoprotein (P), and RNA polymerase (L) [14]. The NiV glycoprotein (NiV-G), is a type II transmembrane glycoprotein that is essential for the virus to enter host cells. A homotrimeric protein, the NiV-G protein is composed of three identical subunits, each of which is made up of two domains: an N-terminal receptor binding domain (RBD) and a C-terminal fusion domain (FD) [15]. The RBD of the NiV-G virus is in charge of identifying and attaching to host cell receptors, which is what permits the virus to enter the cell. The cellular receptor for the NiV has been discovered to be the membrane-bound proteins ephrin-B2 and ephrin-B3, which play a crucial role in cell signalling and development. A recent investigation allowed for this discovery. The attachment of the NiV-G RBD to these receptors is made possible by its high affinity and specificity, and it is a vital stage in the process by which the virus enters host cells [16]. The C-terminal fusion domain (FD) of NiV-G aids in the fusion of the host cell's membrane and viral membrane, allowing the viral entry in cytoplasm of the host cell's. The fusion process is initiated by conformational alterations in the NiV-G protein brought on by receptor contact [17]. The very conserved NiV nucleocapsid protein, also known as NiV-NP, is essential for both viral assembly and replication. The multifunctional NiV-NP participates in viral transcription, RNA synthesis, and genome packing [18]. The ribonucleoprotein complex that the NiV-NP forms serves as a template for viral transcription and replication. By attaching to the virus's RNA genome, this complex is created. In addition, the NiV-NP interacts with the L and P proteins of the viral polymerase complex to promote the synthesis of viral RNA. Additionally, as the virus is being put together, the NiV-NP is in charge of attracting the viral RNA genome into the developing viral particle. According to Wang, et al., this is one of the NiV-NP numerous significant functions in genome packing. It has been proven that antibodies that are focused on NiV-NP can successfully shield animals from NiV infection [19]. According to Weingartl H and colleagues, the immune system also targets the NiV-NP [20]. Immunoinformatic approaches are a powerful method for screening entire pathogen proteomes for potential immunogenic areas in this important race to create a vaccine [21]. For the current investigation, we used an *in silico* methods to create a multiepitope vaccine against NiV.

## 2. Methodology

To identify potential vaccine candidates against NiV, a computational approach was used in the current investigation. In Fig 1 we see a flowchart that summarizes the methodology for using a reverse vaccinology technique to build an epitope-based vaccine.

### 2.1. Retrieval of NiV protein sequence and the analysis of the physiochemical and antigenic properties

From the NCBI database, the attachment glycoprotein (G) and nucleocapid (N) proteins of NiV were obtained.

### 2.2. Cytotoxic T lymphocytes (CTL) and helper T lymphocytes (HTL) epitope prediction

The Immune Epitope Database (IEDB), a comprehensive resource for epitope prediction that includes multiple algorithms and tools, including artificial neural networks and support vector

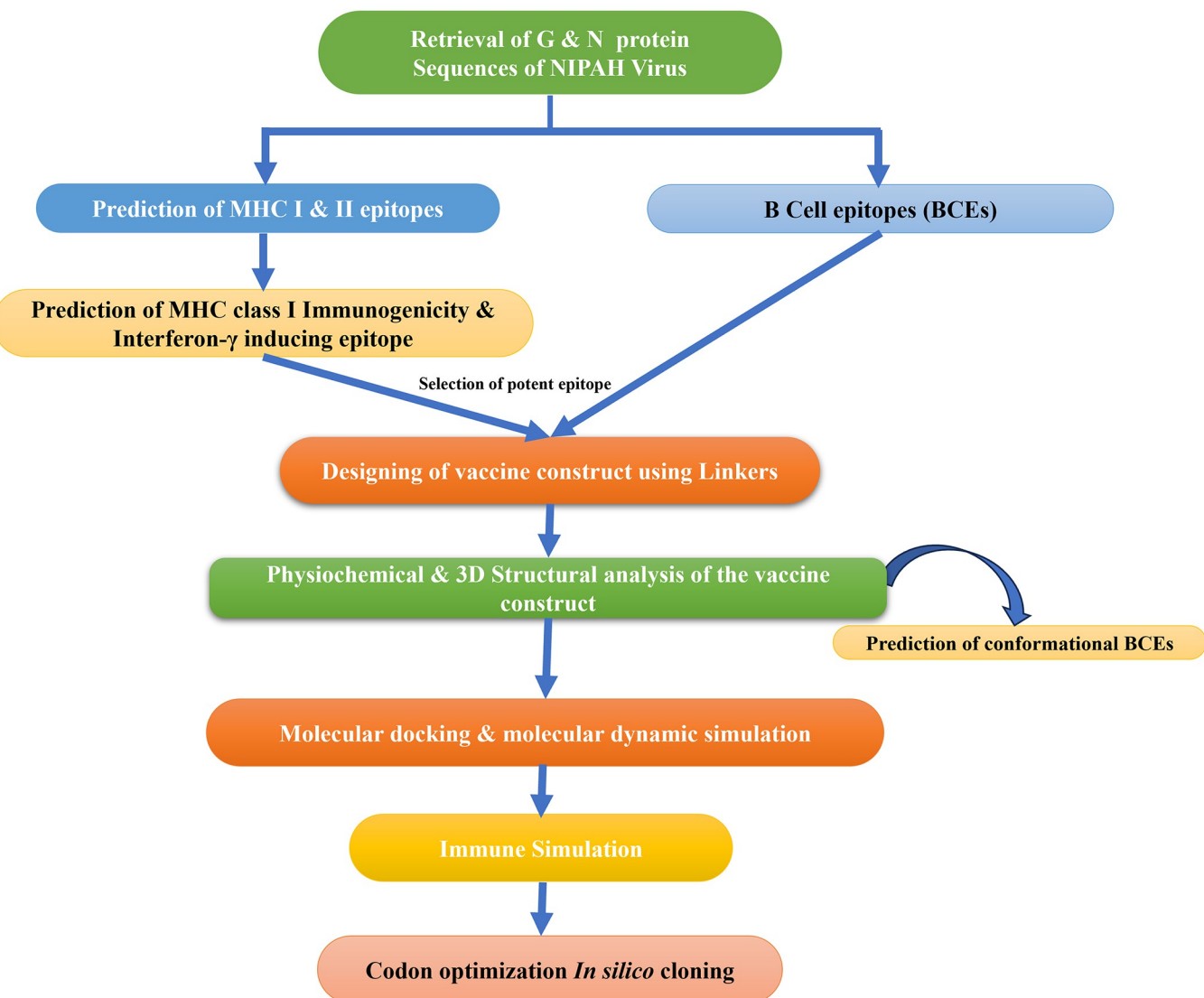

**Fig 1. Flowchart illustration of the methodology that was applied in the present investigation.**

machines. MHC-I and MHC-II epitopes for NiV-G and NiV-NP were predicted using the server's default parameters. Accordingly, the percentile score (lower the percentile value, higher the binding affinity of the predicted epitope) was chosen as <0.5 for both MHC-I and MHC-II alleles.

## 2.3. Immunogenicity prediction of MHC class I epitopes

In order to further analyse the immunogenicity of the MHC-I epitopes identified from the IEDB server, we used the IEDB MHC-I Immunogenicity tool, which examines and verifies exclusively 9-mer peptides [22].

## 2.4. Prediction of Interferon-γ inducing epitope

IFNepitope server was utilized to analyse the predicted epitopes' capacity to induce interferon-gamma (IFN-γ). We predicted IFN-versus-non-IFN-models in this investigation using Motif and SVM hybrid algorithms (accuracy of 82.10%) [23].

## 2.5. Analysis of population coverage

The MHC HLA allele might vary depending on where in the world you are located due to the many different geographical regions. Because of this, conducting an analysis of population coverage is now required before building an effective vaccine construct. To evaluate the global coverage of allele-interaction epitopes of MHC-I and MHC-II, the IEDB population coverage analysis tool was used to conduct a population coverage study [24]. The vast majority of possible prominent epitopes were found in various locations around the world.

## 2.6. Designing and physiochemical analysis of vaccine constructs

*In silico* vaccine constructs based on MHC-I, MHC-II, and B cell epitopes from NiV were created by combining the MHC-I, MHC-II, and B cell epitopes with AAY, GPGPG, and KK linkers, respectively [25,26]. An EAAAK linker along with a cholera toxin adjuvant (Hou, J. et al. 2014; Kim, H. J., et al 2007) is also expected at the N terminus The protein sequence's physicochemical properties, such as molecular weight, aliphatic score, in-vitro half-life instability index, and grand average of hydropathy (GRAVY), were analyzed using the ProtParam program [27].

## 2.7. Prediction of conformational BCEs

The ellipro tool of IEDB was used to predict the conformational BCEs in the vaccine construct. Protein structures were uploaded in PDB format, while all other parameters were left at their default settings. Ellipro works by utilizing the geometric aspects of protein structure, including nearby cluster residues, the overall shape of the protein, and the remaining protrusion index [28].

## 2.8. Allergenicity and antigenicity

Based on the resemblance of potential vaccine candidates to known epitopes, AllerTOP v. 2.0 was used to evaluate the allergenicity of vaccine candidates [29]. VaxiJen 2.0 server was used to evaluate the vaccine candidate's antigenicity [30].

## 2.9. Prediction of secondary structure

Psipred is an internet server that uses the amino acid sequences in a specific manner to produce the secondary structure of the multi-epitope construct (http://bioinf.cs.ucl.ac.uk/psipred/). This is one of the most popular servers for predicting protein secondary structures; it analyzes the PSI-BLAST output using two feed-forward neural networks [31].

## 2.10. Prediction, validation and refinement of the tertiary structure

I-TASSER, a server using the sequence-to-structure-to-function paradigm was used to predict the construct's tertiary structure [25]. The created construct's 3D structure needs to be validated in order to look for any potential mistakes. The Ramachandran plot generated by PDBsum, a visual database that generated the tertiary structure of the vaccine construct, was used to validate it.

## 2.11. B-cell epitope (BCEs) prediction

The primary antigenic determinants recognized by the host immune system are BCEs, which describe a particular antigenic region that binds with the B lymphocytes. A vaccine candidate needs to start an efficient B-cell humoral immune response. By applying default parameters and a score of more than 0.8 for each epitope, we performed the artificial neural network-based ABCpred server analysis uses recurrent neural network for the prediction of B cell epitopes and server has 65.93% accuracy [23].

## 2.12. Disulfide engineering of the vaccine construct

Disulfide engineering is a novel approach to enhance protein structure stability by introducing new disulfide bonds in the target protein through cysteine modification of protein structure residues. Hence, the Disulfide by Design 2 (DbD2) server was employed to detect residue pairings with mutational potential suitable for disulfide engineering [32].

## 2.13. Molecular docking & molecular dynamic simulation

Virus infection triggers an innate immune response through TLRs; in particular, it has been demonstrated that TLR can recognize the viral envelope protein. As a result, the target proteins against which vaccination candidates were developed were TLR3, TLR7, and TLR8. When predicting the binding energy between TLRs and vaccine constructs, the ClusPro 2.0 server was utilized [33]. The experiment was carried out by uploading PDB files of the receptors and ligands into the server and submitting them with the default parameters. The best-scoring vaccine assembly complexed with TLRs was exposed to molecular dynamic simulation by iMODS, a online server after executing the protein-protein molecular docking [34], using the default parameters. Numerous antigen-presenting cells (APCs) including tissue dendritic cells and monocytes include the TLR-3 protein, which can activate the body's cells' unique recognition response to RNA virus infection. Predictions are made about protein complex dynamics simulation in terms of atomic B-factors, eigenvalue variance, deformability, elastic network, and covariance map. The capacity of each of a protein's residues to deform is largely responsible for a protein's deformability. The eigenvalue is associated to the energy needed to distort the specified structure; the lower the eigenvalue value, the easier the complex is to deform. Additionally, according to Sarkar et al, the eigenvalue of the protein complex in question determines the stiffness of its motion [35].

## 2.14. Analysis of the construct's immunological characteristics

The C-immSim immune server was used to model how an actual immune system would react to our final vaccine construct. Immune interactions and immunogenic epitopes can be simulated and predicted using this freely available web-server [36], which is based on the Position-Specific Scoring Framework (PSSM). The program was executed with the default settings, including a vaccination injection without lipopolysaccharide and time step of injection of 1, 42, and 84.

## 2.15. *In silico* cloning optimization of designed vaccine candidate

The multi-epitope vaccine candidate's codons were optimized, and reverse translations were examined using JCAT. The degree of expression of protein was calculated using a codon adaptation index (CAI). Additionally, JCat output included the proportion of GC content [37]. The improved sequence was then cloned using DNASTAR into the pcDNA™3.1/V5-His-TOPO® expression vector.

# 3. Results

For the immunoinformatic investigation, NCBI was contacted for the attachment glycoprotein (G) and nucleocapid (N) proteins of NiV with accession numbers NP 112027.1 and NP 112021.1, respectively.

## 3.1. Cytotoxic T lymphocytes (CTL) and helper T lymphocyte (HTL) epitope prediction

The IEDB server was used to determine particular immunogenic CTL (9-mer) and HTL (15-mer) epitopes using the G and N protein sequences as input. The immunogenicity and percentile rank were used to filter and choose the predicted epitopes, and they were found to be likely antigens. For the various MHC I alleles of the G and N proteins, a total of 24 (Table 1) and 30 (Table 2) predicted epitopes were shown to be quite immunogenic and antigenic. The

**Table 1. MHC-I epitope prediction for the NIPAH glycoprotein.**

| Allele | Start | End | Peptide | Percentile _Rank | Immunogenicity | Antigenicity Score |
|--------|-------|-----|---------|------------------|----------------|--------------------|
| HLA-A*32:01 | 516 | 524 | **RINWISAGV** | 0.74 | 0.24748 | 2.1854 |
| HLA-A*02:06 | | | | 0.93 | | |
| HLA-A*02:03 | | | | 0.32 | | |
| HLA-A*02:01 | | | | 0.99 | | |
| HLA-B*15:01 | 367 | 375 | VGFLVRTEF | 0.94 | 0.20702 | 1.8351 |
| HLA-A*30:01 | 368 | 376 | GFLVRTEFK | 0.96 | 0.27778 | 1.6705 |
| HLA-B*35:01 | 369 | 377 | **FLVRTEFKY** | 0.49 | 0.17033 | 1.4892 |
| HLA-B*15:01 | | | | 0.37 | | |
| HLA-A*30:02 | | | | 0.33 | | |
| HLA-A*26:01 | | | | 0.62 | | |
| HLA-A*01:01 | | | | 0.43 | | |
| HLA-A*11:01 | 100 | 108 | KIGTEIGPK | 0.53 | 0.29496 | 1.3100 |
| HLA-A*03:01 | | | | 0.26 | | |
| HLA-B*08:01 | 476 | 484 | VVNWRNNTV | 0.86 | 0.28232 | 1.2407 |
| HLA-A*68:02 | | | | 0.98 | | |
| HLA-B*51:01 | 101 | 109 | IGTEIGPKV | 0.44 | 0.13949 | 1.2287 |
| HLA-A*68:01 | 164 | 172 | NPLPFREYR | 0.89 | 0.2303 | 1.2044 |
| HLA-A*33:01 | | | | 0.15 | | |

*(Continued)*

**Table 1.** (Continued)

| Allele | Start | End | Peptide | Percentile _Rank | Immunogenicity | Antigenicity Score |
|---|---|---|---|---|---|---|
| HLA-B*35:01 | 364 | 372 | FPAVGFLVR | 0.77 | 0.2122 | 1.2033 |
| HLA-A*33:01 | | | | 0.75 | | |
| HLA-A*26:01 | 119 | 127 | TIPANIGLL | 0.9 | 0.17687 | 1.1538 |
| HLA-B*53:01 | 423 | 431 | **NPKVVFIEI** | 0.48 | 0.29276 | 1.1360 |
| HLA-B*51:01 | | | | 0.02 | | |
| HLA-B*35:01 | | | | 0.98 | | |
| HLA-B*08:01 | | | | 0.11 | | |
| HLA-B*07:02 | | | | 0.33 | | |
| HLA-B*58:01 | 118 | 126 | **ITIPANIGL** | 0.51 | 0.19617 | 1.1090 |
| HLA-B*57:01 | | | | 0.71 | | |
| HLA-A*68:02 | | | | 0.16 | | |
| HLA-A*32:01 | | | | 0.39 | | |
| HLA-A*02:06 | | | | 0.2 | | |
| HLA-A*30:01 | 7 | 15 | KVRFENTTS | 0.37 | 0.28145 | 1.1042 |
| HLA-B*58:01 | 511 | 519 | **AFLIDRINW** | 0.89 | 0.30918 | 0.9509 |
| HLA-B*57:01 | | | | 0.94 | | |
| HLA-A*32:01 | | | | 0.89 | | |
| HLA-A*24:02 | | | | 0.28 | | |
| HLA-A*23:01 | | | | 0.22 | | |
| HLA-B*35:01 | 455 | 463 | QASFSWDTM | 0.7 | 0.15291 | 0.8540 |
| HLA-A*24:02 | 362 | 370 | LYFPAVGFL | 0.12 | 0.2008 | 0.8143 |
| HLA-A*23:01 | | | | 0.09 | | |
| HLA-A*24:02 | 363 | 371 | YFPAVGFLV | 0.39 | 0.20019 | 0.8129 |
| HLA-A*23:01 | | | | 0.41 | | |
| HLA-A*68:02 | 59 | 67 | SIVIIVMNI | 0.94 | 0.1638 | 0.7998 |
| HLA-B*08:01 | 512 | 520 | FLIDRINWI | 0.52 | 0.36516 | 0.7847 |
| HLA-A*02:06 | | | | 0.07 | | |
| HLA-A*02:03 | | | | 0.02 | | |
| HLA-A*02:01 | | | | 0.03 | | |
| HLA-B*15:01 | 361 | 369 | TLYFPAVGF | 0.25 | 0.19727 | 0.7663 |
| HLA-A*32:01 | | | | 0.13 | | |
| HLA-A*26:01 | | | | 0.52 | | |
| HLA-A*23:01 | | | | 0.95 | | |
| HLA-B*15:01 | 248 | 256 | RIIGVGEVL | 0.6 | 0.25802 | 0.7453 |
| HLA-B*07:02 | | | | 0.98 | | |
| HLA-A*32:01 | | | | 0.3 | | |
| HLA-A*02:06 | | | | 0.6 | | |
| HLA-A*68:02 | 254 | 262 | EVLDRGDEV | 0.1 | 0.17824 | 0.7411 |
| HLA-A*26:01 | | | | 0.57 | | |
| HLA-B*35:01 | 223 | 231 | **AMDEGYFAY** | 0.31 | 0.25913 | 0.7221 |
| HLA-B*15:01 | | | | 0.25 | | |
| HLA-A*32:01 | | | | 0.39 | | |
| HLA-A*30:02 | | | | 0.05 | | |
| HLA-A*26:01 | | | | 0.56 | | |
| HLA-A*02:06 | | | | 0.96 | | |
| HLA-A*01:01 | | | | 0.02 | | |
| HLA-B*08:01 | 244 | 252 | VSKQRIIGV | 0.35 | 0.12124 | 0.7018 |
| HLA-A*30:01 | | | | 0.31 | | |

**Table 2. MHC-I epitope prediction for the nucleocapsid protein.**

| Allele | Start | End | Peptide | Score | Percentile Rank | Immunogenicity | Antigenicity Score |
|--------|-------|-----|---------|-------|-----------------|----------------|--------------------|
| HLA-A*03:01 | 26 | 34 | **AATATLTTK** | 0.517945 | 0.32 | 0.13477 | 1.1018 |
| HLA-A*11:01 | | | | 0.746321 | 0.1 | | |
| HLA-A*30:01 | | | | 0.397764 | 0.26 | | |
| HLA-A*68:02 | 43 | 51 | NSPELRWEL | 0.27533 | 0.47 | 0.38051 | 0.7535 |
| HLA-B*40:01 | 45 | 53 | PELRWELTL | 0.23485 | 0.58 | 0.37175 | 1.4427 |
| HLA-A*26:01 | 46 | 54 | ELRWELTLF | 0.373933 | 0.19 | 0.35303 | 1.3582 |
| HLA-B*15:01 | | | | 0.294962 | 0.65 | | |
| HLA-A*24:02 | 48 | 56 | RWELTLFAL | 0.087344 | 0.92 | 0.17036 | 0.9332 |
| HLA-A*32:01 | 69 | 77 | KVGAAFTLI | 0.229765 | 0.33 | 0.22495 | 0.8481 |
| HLA-B*35:01 | 71 | 79 | GAAFTLISM | 0.121495 | 0.9 | 0.17352 | 0.7847 |
| HLA-A*23:01 | 79 | 87 | **MYSERPGAL** | 0.124381 | 0.66 | 0.13847 | 0.7632 |
| HLA-A*24:02 | | | | 0.228902 | 0.43 | 0.13847 | |
| HLA-B*08:01 | | | | 0.365206 | 0.28 | 0.13847 | |
| HLA-A*01:01 | 80 | 88 | YSERPGALI | 0.104778 | 0.91 | 0.13222 | 0.8029 |
| HLA-B*07:02 | 83 | 91 | **RPGALIRSL** | 0.981392 | 0.01 | 0.11187 | 0.6313 |
| HLA-B*08:01 | | | | 0.299036 | 0.35 | 0.11187 | |
| HLA-B*53:01 | | | | 0.07791 | 0.92 | 0.11187 | |
| HLA-B*51:01 | 94 | 102 | DPDIEAVII | 0.410317 | 0.29 | 0.38805 | 1.0875 |
| HLA-B*53:01 | | | | 0.146884 | 0.56 | | |
| HLA-A*26:01 | 99 | 107 | AVIIDVGSM | 0.224464 | 0.33 | 0.16952 | 0.6558 |
| HLA-A*24:02 | 150 | 158 | AYGLRITDM | 0.103734 | 0.83 | 0.22124 | 1.9333 |
| HLA-A*68:02 | 162 | 170 | VSAVITIEA | 0.153048 | 0.83 | 0.3912 | 0.8755 |
| HLA-A*02:06 | 164 | 172 | **AVITIEAQI** | 0.377104 | 0.42 | 0.27145 | 0.9413 |
| HLA-A*32:01 | | | | 0.089754 | 0.81 | | |
| HLA-A*68:02 | | | | 0.331094 | 0.36 | | |
| HLA-A*30:01 | 202 | 210 | RVNPFFALT | 0.322416 | 0.38 | 0.23748 | 1.6784 |
| HLA-B*08:01 | 243 | 251 | SAKGRAVEI | 0.720362 | 0.06 | 0.14467 | 1.1588 |
| HLA-B*51:01 | | | | 0.316318 | 0.44 | 0.14467 | |
| HLA-A*32:01 | 265 | 273 | AGFFATIRF | 0.114977 | 0.66 | 0.373 | 0.6086 |
| HLA-A*23:01 | 267 | 275 | FFATIRFGL | 0.09218 | 0.84 | 0.34868 | 0.9768 |
| HLA-A*24:02 | | | | 0.078825 | 0.98 | 0.34868 | |
| HLA-A*31:01 | 270 | 278 | TIRFGLETR | 0.673279 | 0.15 | 0.26434 | 1.6965 |
| HLA-A*33:01 | | | | 0.563756 | 0.12 | | |
| HLA-A*68:01 | | | | 0.745208 | 0.26 | | 1.6965 |
| HLA-A*30:02 | 271 | 279 | IRFGLETRY | 0.35508 | 0.33 | 0.21855 | 2.2023 |
| HLA-B*08:01 | 274 | 282 | GLETRYPAL | 0.326862 | 0.32 | 0.13198 | 1.2908 |
| HLA-A*01:01 | 319 | 327 | QTKFAPGGY | 0.263989 | 0.43 | 0.12386 | 1.4435 |
| HLA-A*26:01 | | | | 0.541353 | 0.1 | | |
| HLA-A*30:02 | | | | 0.530767 | 0.15 | | |
| HLA-B*15:01 | | | | 0.298431 | 0.64 | | |
| HLA-B*07:02 | 327 | 335 | **YPLLWSFAM** | 0.403738 | 0.34 | 0.16687 | 0.8671 |
| HLA-B*08:01 | | | | 0.199392 | 0.59 | | |
| HLA-B*35:01 | | | | 0.909678 | 0.04 | | |
| HLA-B*51:01 | | | | 0.403517 | 0.3 | | |
| HLA-B*53:01 | | | | 0.486961 | 0.14 | | |

(*Continued*)

**Table 2.** (Continued)

| Allele | Start | End | Peptide | Score | Percentile Rank | Immunogenicity | Antigenicity Score |
|---|---|---|---|---|---|---|---|
| HLA-A*02:06 | 333 | 341 | **FAMGVATTI** | 0.423595 | 0.35 | 0.10957 | 0.7167 |
| HLA-A*68:02 | | | | 0.4186 | 0.26 | | |
| HLA-B*35:01 | | | | 0.274233 | 0.46 | | |
| HLA-B*51:01 | | | | 0.899183 | 0.02 | | |
| HLA-B*53:01 | | | | 0.314859 | 0.27 | | |
| HLA-A*31:01 | 344 | 352 | SMGALNINR | 0.587371 | 0.25 | 0.14202 | 1.0967 |
| HLA-A*33:01 | | | | 0.179058 | 0.82 | | |
| HLA-A*30:02 | 346 | 354 | GALNINRGY | 0.424699 | 0.24 | 0.17688 | 1.1678 |
| HLA-B*15:01 | | | | 0.216611 | 0.87 | | |
| HLA-A*03:01 | 357 | 365 | PMYFRLGQK | 0.213863 | 0.89 | 0.11748 | 1.6454 |
| HLA-A*68:02 | 412 | 420 | EAKFAAGGV | 0.173742 | 0.75 | 0.17113 | 1.1882 |
| HLA-A*31:01 | 521 | 529 | AQNDLDFVR | 0.442338 | 0.45 | 0.15322 | 1.0682 |

several CTL epitopes that were predicted exhibited strong binding affinity to multiple MHC I alleles. Thirteen HTL epitopes were G protein epitopes (Table 3), and 28 HTL epitopes for N protein, all predicted epiotpes were found to be IFN-inducing epitopes (Table 4).

## 3.2. Population coverage

It was anticipated that the vaccine that would be created would be able to protect a significant number of people all over the world. The evaluation of the distribution of CTL and HTL epitope alleles that were employed in the development of the vaccine construct was carried out with the assistance of the IEDB population coverage tool, which can be found at http://tools.iedb.org/population/. Fig 2 shows that the MHC I and MHC II epitopes that are employed for population coverage study may cover 96.76% of the world's population.

Population coverage of the selected epitopes was 84.51% in Central Africa, 90.22% in East Africa, 89.16% in North Africa, 93.63% in South Africa, and 94.25% in West Africa, respectively. The percentages for East Asia, Northeast Asia, South Asia, Southeast Asia, and Southwest Asia were 96.54%, 92.1%, 89.45%, 92.1%, and 88.6%, respectively. Fig 2 shows that the

**Table 3. MHC-II epitope prediction for the NIPAH glycoprotein.**

| Allele | Start | End | Peptide | Percentile Rank | IFN γ | |
|---|---|---|---|---|---|---|
| HLA-DRB3*01:01 | 510 | 524 | DAFLIDRINWISAGV | 0.22 | POSITIVE | 0.390779 |
| HLA-DPA1*02:01/ DPB1*05:01 | 364 | 378 | FPAVGFLVRTEFKYN | 0.8 | POSITIVE | 0.619056 |
| HLA-DRB1*03:01 | 537 | 551 | FTVFKDNEILYRAQL | 0.54 | POSITIVE | 0.21947 |
| **HLA-DRB3*01:01** | **453** | **467** | **FYQASFSWDTMIKFG** | **0.56** | **POSITIVE** | **1** |
| **HLA-DRB3*01:01** | 506 | 520 | **GVYNDAFLIDRINWI** | 0.2 | POSITIVE | 0.188534 |
| HLA-DRB1*03:01 | 534 | 548 | NPVFTVFKDNEILYR | 0.54 | POSITIVE | 0.118736 |
| HLA-DRB3*02:02 | 474 | 488 | PLVVNWRNNTVISRP | 1 | POSITIVE | 0.524562 |
| HLA-DRB1*03:01 | 535 | 549 | PVFTVFKDNEILYRA | 0.54 | POSITIVE | 0.497109 |
| HLA-DRB1*03:01 | 536 | 550 | VFTVFKDNEILYRAQ | 0.54 | POSITIVE | 0.379089 |
| **HLA-DRB3*01:01** | **452** | **466** | **VFYQASFSWDTMIKF** | **0.46** | **POSITIVE** | **1** |
| HLA-DRB3*01:01 | 507 | 521 | VYNDAFLIDRINWIS | 0.15 | POSITIVE | 0.242859 |
| HLA-DRB3*01:01 | 508 | 522 | YNDAFLIDRINWISA | 0.15 | POSITIVE | 0.305007 |
| HLA-DRB3*01:01 | 454 | 468 | YQASFSWDTMIKFGD | 0.07 | POSITIVE | 1 |

**Table 4. MHC-II epitope prediction for the nucleocapsid protein.**

| Allele | Start | End | Peptide | Percentile Rank | IFN-γ | |
|---|---|---|---|---|---|---|
| HLA-DQA1*04:01/DQB1*04:02 | 385 | 399 | SSDQVAELAAAVQET | 0.07 | POSITIVE | 0.166795 |
| HLA-DRB1*09:01 | 331 | 345 | WSFAMGVATTIDRSM | 0.11 | POSITIVE | 0.166795 |
| HLA-DQA1*01:01/DQB1*05:01 | 45 | 59 | PELRWELTLFALDVI | 0.12 | POSITIVE | 0.208659 |
| **HLA-DPA1*01:03/DPB1*04:01** | 265 | 279 | **AGFFATIRFGLETRY** | 0.14 | POSITIVE | 0.785572 |
| HLA-DPA1*01:03/DPB1*04:01 | 264 | 278 | MAGFFATIRFGLETR | 0.14 | POSITIVE | 0.785572 |
| **HLA-DRB1*04:05** | 83 | 97 | **RPGALIRSLLNDPDI** | 0.14 | POSITIVE | 0.167126 |
| HLA-DPA1*01:03/DPB1*04:01 | 263 | 277 | GMAGFFATIRFGLET | 0.15 | POSITIVE | 0.363825 |
| HLA-DRB1*13:02 | 247 | 261 | RAVEIISDIGNYVEE | 0.21 | POSITIVE | 0.207261 |
| HLA-DRB1*13:02 | 249 | 263 | VEIISDIGNYVEETG | 0.21 | POSITIVE | 0.027157 |
| HLA-DQA1*04:01/DQB1*04:02 | 384 | 398 | LSSDQVAELAAAVQE | 0.26 | POSITIVE | 0.049607 |
| HLA-DPA1*01:03/DPB1*04:01 | 46 | 60 | ELRWELTLFALDVIR | 0.33 | POSITIVE | 0.05826 |
| HLA-DPA1*02:01/DPB1*14:01 | 32 | 46 | TTKIRIFVPATNSPE | 0.33 | POSITIVE | 0.15079 |
| **HLA-DRB1*04:05** | 124 | 138 | **EMEGLMRILKTARDS** | 0.35 | POSITIVE | 0.517311 |
| HLA-DRB1*04:01 | 285 | 299 | NEFQSDLNTIKSLML | 0.46 | POSITIVE | 0.126975 |
| HLA-DQA1*03:01/DQB1*03:02 | 388 | 402 | QVAELAAAVQETSAG | 0.47 | POSITIVE | 0.141264 |
| HLA-DRB3*01:01 | 247 | 261 | RAVEIISDIGNYVEE | 0.5 | POSITIVE | 0.40246 |
| HLA-DRB3*01:01 | 246 | 260 | GRAVEIISDIGNYVE | 0.56 | POSITIVE | 0.583983 |
| HLA-DQA1*03:01/DQB1*03:02 | 384 | 398 | LSSDQVAELAAAVQE | 0.56 | POSITIVE | 0.583983 |
| HLA-DRB1*04:05 | 36 | 50 | RIFVPATNSPELRWE | 0.63 | POSITIVE | 0.812931 |
| HLA-DRB1*15:01 | 288 | 302 | QSDLNTIKSLMLLYR | 0.64 | POSITIVE | 0.812931 |
| HLA-DRB1*01:01 | 472 | 486 | LTNSLLNLRSRLAAK | 0.67 | POSITIVE | 0.09411 |
| HLA-DRB1*01:01 | 473 | 487 | TNSLLNLRSRLAAKA | 0.67 | POSITIVE | 0.309278 |
| HLA-DPA1*02:01/DPB1*14:01 | 444 | 458 | VTFKREMSISSLANS | 0.73 | POSITIVE | 0.055074 |
| HLA-DRB1*15:01 | 473 | 487 | TNSLLNLRSRLAAKA | 0.78 | POSITIVE | 0.264098 |
| **HLA-DRB1*13:02** | 372 | 386 | **GGIDQNMANRLGLSS** | 0.81 | POSITIVE | 0.395926 |
| HLA-DRB1*15:01 | 474 | 488 | NSLLNLRSRLAAKAA | 0.84 | POSITIVE | 0.130847 |
| HLA-DQA1*05:01/DQB1*03:01 | 413 | 427 | AKFAAGGVLIGGSDQ | 0.85 | POSITIVE | 0.130847 |
| HLA-DRB1*01:01 | 471 | 485 | RLTNSLLNLRSRLAA | 0.91 | POSITIVE | 0.375392 |

percentages were as follows: 98.86% in Europe, 97.13% in the West Indies, 12.45% in Central America, 97.19% in North America, and 75.59% in South America. This proportion was 93.9 percent in Oceania. These findings show that the predictable epitopes are appropriate for use in vaccine construction and cover a wide spectrum of the human population.

## 3.3. Identification and prediction of B cell epitopes (BCEs)

The ABCpred server was utilized in order to make the prediction of the B cell epitopes using a cutoff value of 0.8. There were a total of 26 and 24 B cell epitopes predicted for the Nipah virus's G (Table 5) and N (Table 6) proteins, respectively.

## 3.4. Preparation of vaccine construct and physiochemical properties

Using the AAY, GPGPG, and KK linkers, the most promising MHC-I, MHC-II, and B cell epitopes were combined to create the vaccine design. The 124 amino acid long Cholera Toxin B (CTB), a potent adjuvant was used to enhance the immune response to co-administered antigens. As an adjuvant, CTB has been found to increase the production of antigen-specific antibodies and stimulate the activation of T-cells, leading to a more robust and effective immune

| MHC class | Coverage | Average hit | PC90 |
|---|---|---|---|
| combined | 96.76% | 4.21 | 1.52 |

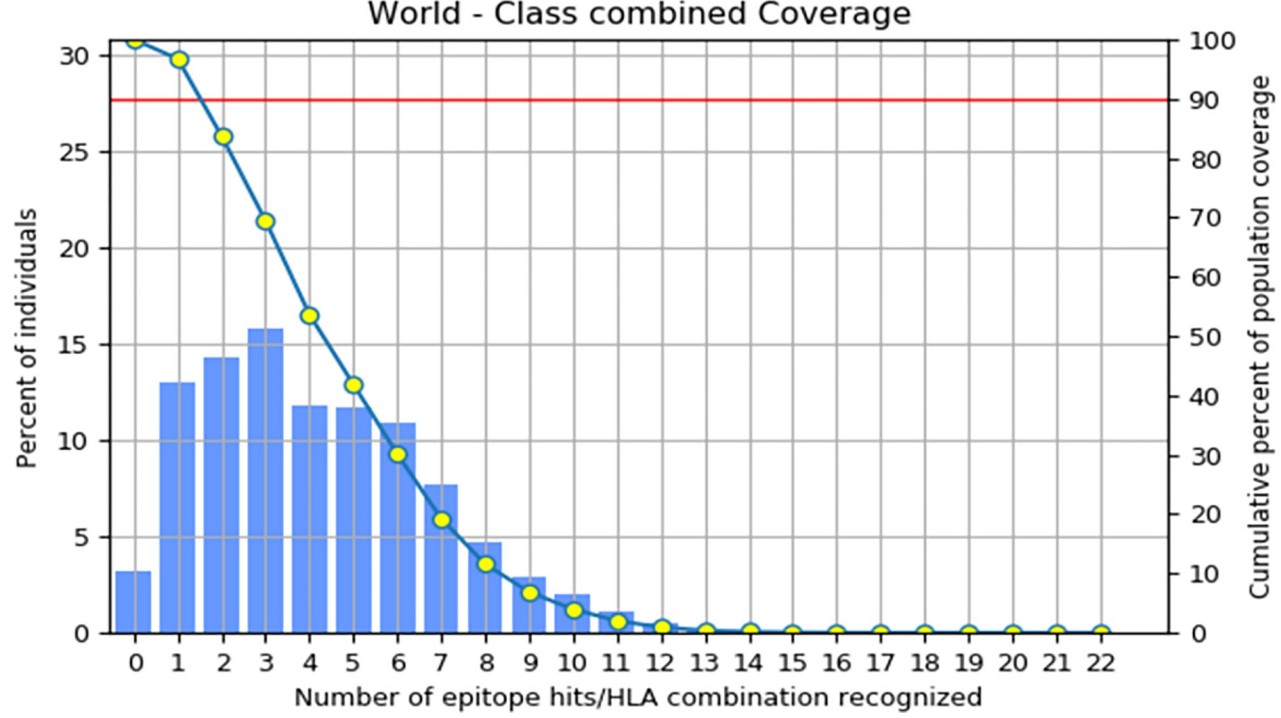

**Fig 2. Shows the global coverage for the selected CTL & HTL epitopes for the vaccine construct.**

**Table 5. List of linear B-cell epitopes with their location and score by using the ABCpred server in NIPAH glycoprotein.**

| Rank | Sequence | Start position | Score |
|---|---|---|---|
| 1. | **SFSWDTMIKFGDVLTV** | 457 | 0.93 |
| 2. | **YVLCAVSTVGDPILNS** | 292 | 0.92 |
| 3. | **TNVWTPPNPNTVYHCS** | 268 | 0.92 |
| 4. | **KKVRFENTTSDKGKIP** | 6 | 0.9 |
| 5. | **KVIKSYYGTMDIKKIN** | 23 | 0.9 |
| 6. | LVEIYDTGDNVIRPKL | 577 | 0.88 |
| 7. | DVLTVNPLVVNWRNNT | 468 | 0.88 |
| 8. | RGDEVPSLFMTNVWTP | 258 | 0.88 |
| 9. | GTCITDPLLAMDEGYF | 214 | 0.87 |
| 10. | VSLIDTSSTITIPANI | 109 | 0.87 |
| 11. | HCSAVYNNEFYYVLCA | 281 | 0.86 |
| 12. | PNNICLQKTSNQILKP | 185 | 0.85 |
| 13. | NISCPNPLPFREYRPQ | 159 | 0.85 |
| 14. | GSKISQSTASINENVN | 128 | 0.85 |
| 15. | SKPENCRLSMGIRPNS | 390 | 0.84 |
| 16. | KVMPYGPSGIKQGDTL | 347 | 0.84 |
| 17. | INWISAGVFLDSNQTA | 517 | 0.83 |

*(Continued)*

**Table 5.** (Continued)

| Rank | Sequence | Start position | Score |
|------|----------|----------------|-------|
| 18. | AFLIDRINWISAGVFL | 511 | 0.83 |
| 19. | YRPQTEGVSNLVGLPN | 171 | 0.83 |
| 20. | NCPITKCQYSKPENCR | 381 | 0.82 |
| 21. | LERIGSCSRGVSKQRI | 234 | 0.82 |
| 22. | VVFIEISDQRLSIGSP | 426 | 0.81 |
| 23. | ALRSIEKGRYDKVMPY | 336 | 0.81 |
| 24. | NSTYWSGSLMMTRLAV | 306 | 0.81 |
| 25. | YRAQLASEDTNAQKTI | 547 | 0.8 |
| 26. | PEICWEGVYNDAFLID | 500 | 0.8 |

response. The final 572 amino acid peptide vaccine design (Fig 3) was created. The physio-chemical parameters of the generated 572 amino acid long vaccine construct were determined using the protparam server, which revealed a molecular weight of 63274.29 and a theoretical pI of 9.79, indicating its basic nature and computed instability index of 28.56 indicates that the construct is stable. The aliphatic index was 78.08, which indicates that the substance is thermo-stable throughout a range of temperatures. Additionally, the grand average of hydropathicity (GRAVY) was -0.218, which shows that the vaccine design is hydrophobic. Protein-Sol server score of 0.608 indicated that our vaccine construct had a good solubility rate.

**Table 6. List of linear B-cell epitopes with their location and score by using the ABCpred server in NIPAH Nucleocapsid protein.**

| Rank | Sequence | Start position | Score |
|------|----------|----------------|-------|
| 1. | **ETRRWAKYVQQKRVNP** | 190 | 0.94 |
| 2. | **ALTQQWLTEMRNLLSQ** | 208 | 0.92 |
| 3. | **AISNRTQGESEKKNNQ** | 501 | 0.91 |
| 4. | **MRILKTARDSSKGKTP** | 129 | 0.9 |
| 5. | MGVATTIDRSMGALNI | 335 | 0.89 |
| 6. | EAVIIDVGSMVNGIPV | 98 | 0.86 |
| 7. | NGIPVMERRGDKAQEE | 109 | 0.86 |
| 8. | DQDIDEGEEPIEQSGR | 426 | 0.86 |
| 9. | AKAAKEAASSNATDDP | 485 | 0.86 |
| 10. | IRSPSAAESMKVGAAF | 59 | 0.85 |
| 11. | TLISMYSERPGALIRS | 75 | 0.84 |
| 12. | GNYVEETGMAGFFATI | 256 | 0.84 |
| 13. | NRGYLEPMYFRLGQKS | 351 | 0.84 |
| 14. | AVEIISDIGNYVEETG | 248 | 0.83 |
| 15. | FATIRFGLETRYPALA | 268 | 0.83 |
| 16. | YREIGPRAPYMVLLEE | 301 | 0.82 |
| 17. | LLEESIQTKFAPGGYP | 313 | 0.82 |
| 18. | FRSYQSKLGRDGRASA | 11 | 0.81 |
| 19. | TGMAGFFATIRFGLET | 262 | 0.81 |
| 20. | EQSGRQSVTFKREMSI | 437 | 0.81 |
| 21. | RYPALALNEFQSDLNT | 278 | 0.8 |
| 22. | FRLGQKSARHHAGGID | 360 | 0.8 |
| 23. | AAAVQETSAGRQESNV | 393 | 0.8 |
| 24. | VLIGGSDQDIDEGEEP | 420 | 0.8 |

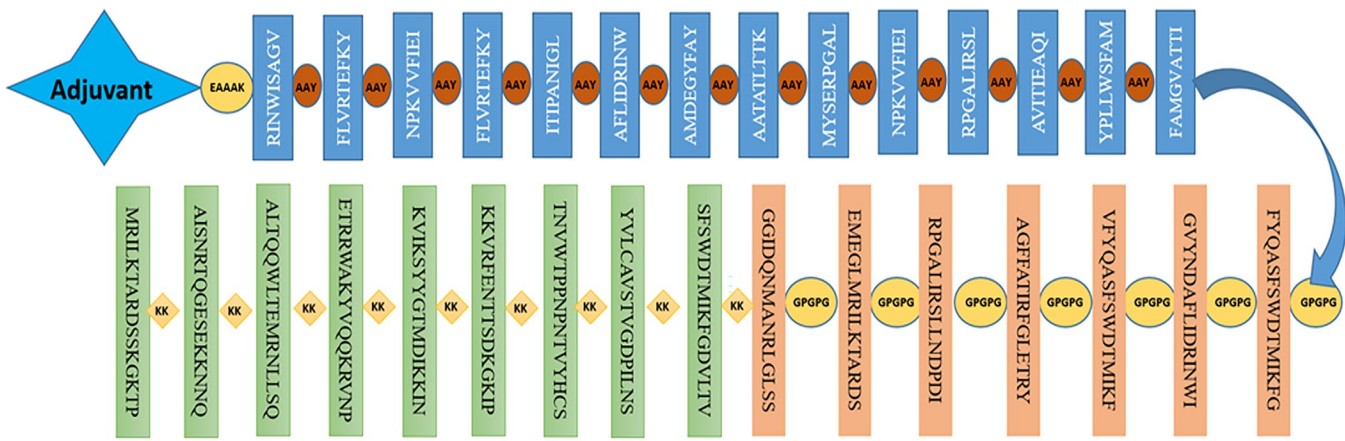

**Fig 3. Graphical representation of the prepared vaccine construct.**

### 3.5. Secondary structure prediction

An accurate prediction of the secondary structure of the NiV vaccine was achieved using PSIPRED. Fig 4 shows the residues that make up the alpha helix to be represented in a pink color, the residues that make up the beta strand to be represented in a yellow color, and the residues that make up the coil to be represented in a gray color. Alpha helix is represented by the pink color, β strand by the yellow color, and coil by the gray color

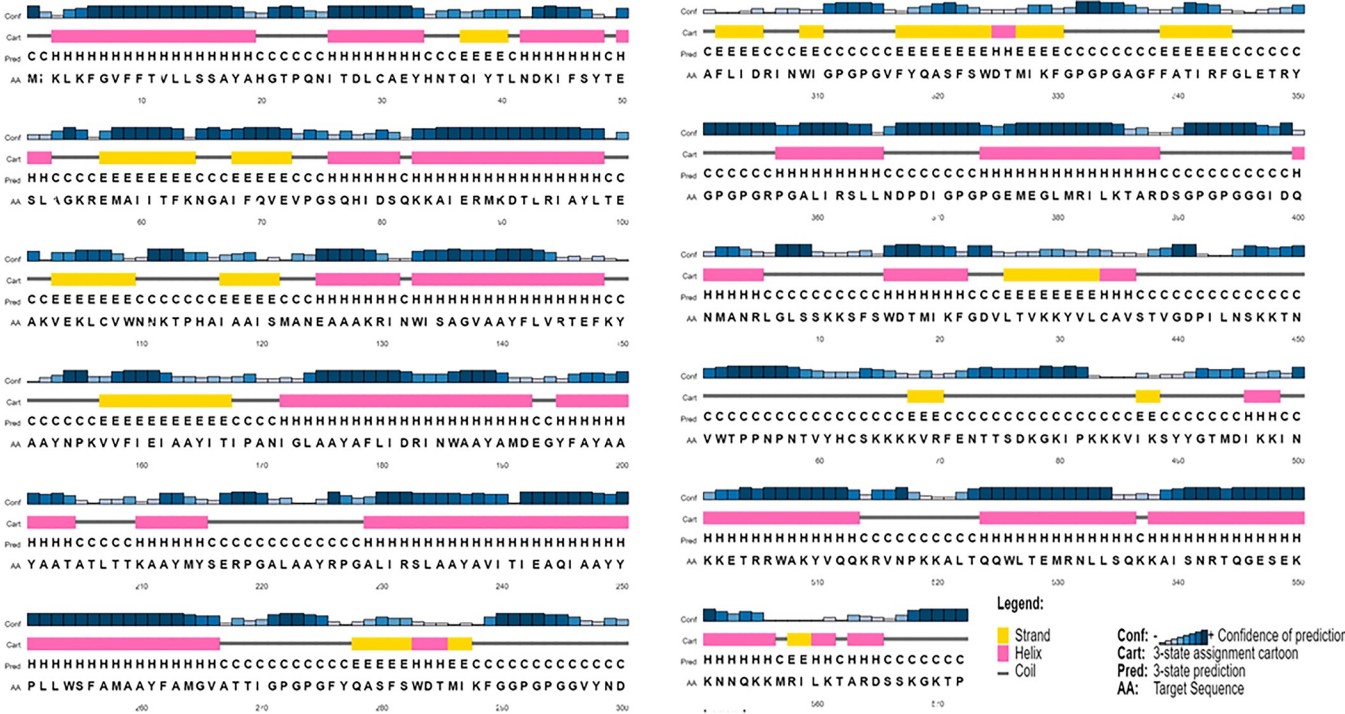

**Fig 4. Graphical representation of the secondary structure of the subunit multi-epitopes vaccine construct predicted by the PSIPRED server.**

### 3.6. Allergenicity and antigenicity

It was hypothesized that the multiepitope vaccine design would not trigger allergic reactions when run through the AllerTOPv2.0 algorithm. Our multiepitope construct received a predicted antigenicity score of 0.5512 from Vaxijen v2.0. This score was above the threshold value of 0.4%, suggesting that the construct possessed antigenic characteristics.

### 3.7. Prediction, validation and refinement of the tertiary structure

I-TASSER predicted a total of 5 models utilizing 10 threading templates with Z-scores, and the models were chosen due to their high confidence C values, which vary from -0.63 to -4.18. Five models were created using the GalaxyRefine server to refine the chimeric vaccine's 3D structure (Fig 5A). Model 1 was chosen for molecular dynamic (MD) simulation based on its model quality score, which included GDT-HA (0.9318) and RMSD (0.480) (Fig 5B). The 3D structure was validated to look for quality issues and probable mistakes. The 3D structure analysis performed by ProSA-web after the MD simulation shows that the structure had a Z-score of 7.52 or less (Fig 5C). According to the Ramachandran plot generated by PROCHECK, 90.3% of the residues are in the regions with the most favorable conditions, 8.3% are in the further permitted regions, and 0.2% are in the liberally permitted regions with no residue in the prohibited regions (Fig 5D).

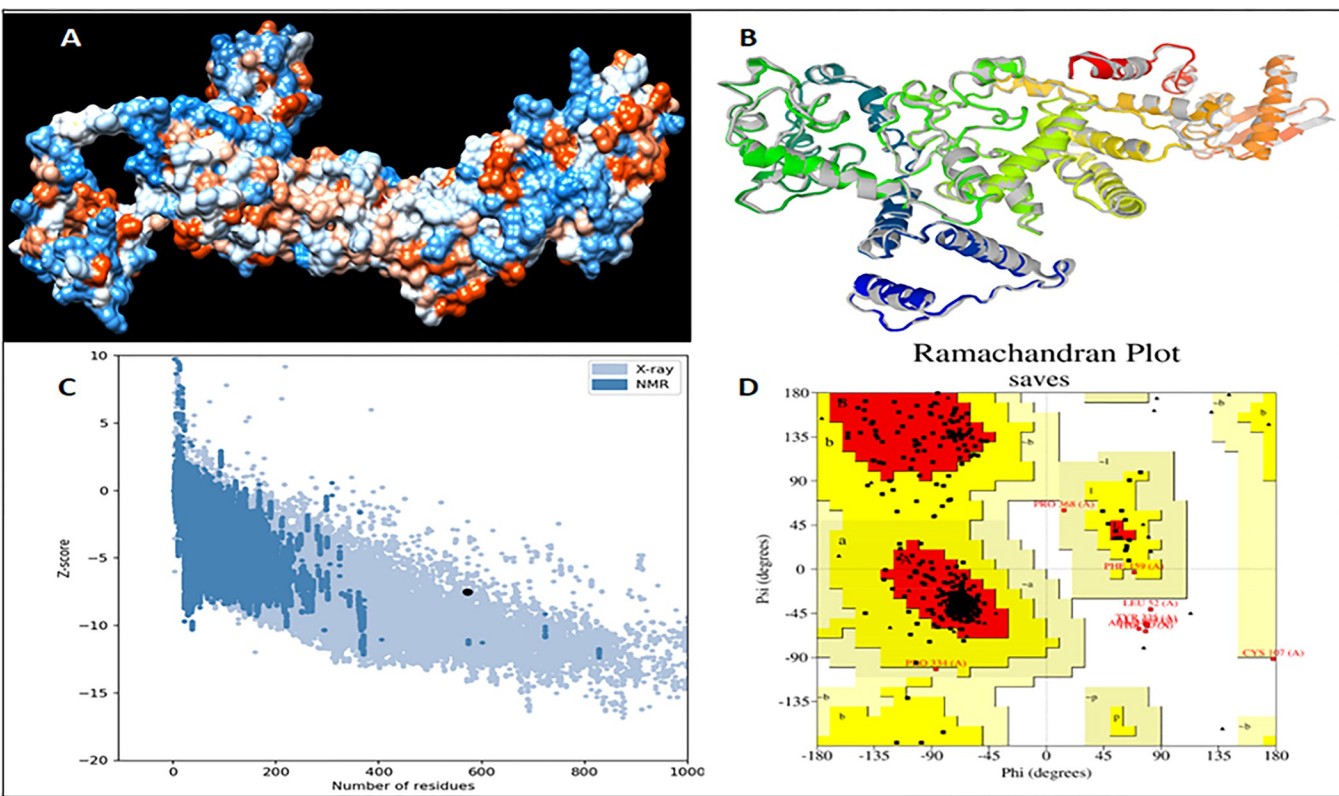

**Fig 5.** Structure prediction, refinement and validation of the 3D structure of the prepared vaccine construct (a) predicted 3d structure by I-TASSER (b) refinement of the 3D structure of the chimeric vaccine by using galaxyRefine server (c) ProSA Z-score plot shows a -7.52 score in the range of conformation of the predicted protein (d) PROCHECK's Ramachandran plot illustrates that the residues are placed in the allowed and favored regions.

**Table 7. List of discontinuous epitopes and their predicted scores.**

| No. | Residues | Number of residues | Score |
|---|---|---|---|
| 1 | A:M1, A:I2, A:K3, A:L4, A:K5 | 5 | 0.92 |
| 2 | A:Q512, A:R515, A:V516, A:N517, A:P518, A:K519, A:K520, A:A521, A:L522, A:T523, A:Q524, A:Q525, A:W526, A:L527, A:T528, A:E529, A:M530, A:R531, A:N532, A:L533, A:L534, A:S535, A:Q536, A:K537, A:K538, A:G546, A:E547, A:S548, A:E549, A:K550, A:K551, A:N552, A:N553, A:Q554, A:K555, A:K556, A:M557, A:R558, A:I559, A:L560, A:K561, A:T562, A:A563, A:R564, A:D565, A:S566, A:S567, A:K568, A:G569, A:K570, A:T571, A:P572 | 52 | 0.844 |
| 3 | A:E378, A:G379, A:M381, A:R382, A:I383, A:L384, A:K385, A:T386, A:A387, A:R388, A:D389, A:S390, A:G391, A:P392, A:G393, A:P394, A:G395, A:G396, A:G397, A:I398, A:D399, A:Q400, A:N401, A:M402, A:A403, A:N404, A:R405, A:L406, A:G407, A:L408, A:S409, A:S410, A:K411, A:K412, A:S413, A:F414, A:S415, A:W416, A:D417, A:T418, A:M419, A:I420, A:K421, A:F422, A:G423, A:D424, A:V425, A:L426, A:T427, A:V428, A:K429, A:L433, A:C434, A:S437, A:T438, A:V439, A:G440, A:D441, A:P442, A:I443, A:L444, A:N445, A:S446, A:K447, A:K448, A:T449, A:N450, A:V451, A:W452, A:T453, A:P454, A:P455, A:N456, A:P457, A:N458, A:T459, A:V460, A:Y461, A:H462, A:C463, A:S464, A:K465, A:K466, A:K467, A:K468, A:V469, A:R470, A:F471, A:E472, A:N473, A:T474, A:T475, A:S476, A:D477, A:K478, A:G479, A:K480, A:I481, A:P482, A:K483, A:K484, A:K485, A:V486, A:I487, A:K488, A:S489, A:Y490 | 107 | 0.724 |
| 4 | A:F6, A:G7, A:V8, A:F9, A:F10, A:T11, A:V12, A:L13, A:L14, A:S15, A:S16, A:A17, A:Y18, A:A19, A:H20, A:G21, A:T22, A:P23, A:Q24, A:N25, A:I26, A:T27, A:D28, A:L29, A:C30, A:A31, A:E32, A:Y33, A:H34, A:N35, A:T36, A:Q37, A:I38, A:Y39, A:T40, A:L41, A:N42, A:D43, A:K44, A:I45, A:F46, A:S47, A:Y48, A:T49, A:E50, A:S51, A:L52, A:A53, A:G54, A:K55, A:E57, A:M58, A:A59, A:I60, A:I61, A:T62, A:F63, A:K64, A:N65, A:G66, A:A67, A:I68, A:F69, A:Q70, A:V71, A:E72, A:V73, A:P74, A:G75, A:S76, A:Q77, A:H78, A:I79, A:D80, A:S81, A:Q82, A:K83, A:K84, A:A85, A:I86, A:E87, A:R88, A:M89, A:D91, A:T92, A:L93, A:R94, A:I95, A:A96, A:Y97, A:L98, A:T99, A:E100, A:A101, A:K102, A:V103, A:E104, A:K105, A:L106, A:C107, A:V108, A:N111, A:K112, A:I120, A:A123, A:N124, A:A126, A:A127, A:A128, A:K129, A:R130, A:I131, A:N132, A:I134, A:S135, A:A136, A:G137, A:A140, A:V158, A:F159, A:E161, A:I162, A:A163, A:A164, A:L180 | 125 | 0.684 |
| 5 | A:Q513, A:K514, A:I540, A:S541, A:N542, A:Q545 | 6 | 0.507 |

### 3.8. Conformational BCEs & disulfide engineering of the vaccine construct

A total of 5 conformational epitope has been predicted in the vaccine construct and have the higher score of 0.5 (Table 7). The vaccine construct's model identifies 32 residue pairs that may potentially create a disulfide bond, as predicted by the Disulfide by Design 2.13 server (Table 8). As per study by Craig DB, et al., for the formation of disulfide bonds the χ3 angle lies between -87° and +97° and energy value less than 2.2 kcal/mol, on the basis of the this only two amino acid pair (335-GLY-373-GLY & 24-GLN-28-ASP) were form the disulfide bond in the designed construct (Table 8).

### 3.9. Molecular docking of vaccine construct with receptors

Through molecular docking, the stability and binding affinity of a docked complex between a ligand and receptor molecule can be examined. For pathogen identification and immune response, toll-like receptor is a crucial human protein. In light of this, we select TLR3, 7, and 8 as the immunological receptors for the molecular docking. The revised 3D model of our finished vaccination is molecularly docked with the TLR3, 7, and 8 immunological receptors using the ClusPro 2.0 server (Kozakov D et al., 2017). The multi-epitope vaccination and TLRs (3, 7 and 8) were molecularly docked using the ClusPro 2.0 server, which created docked complexes with various cluster members and the lowest energy. All docked complexes were compared, and the Clusters with the lowest energy scores were chosen. The residues involved in the interaction between TLR3 (Fig 6A), TLR 7 (Fig 6B) and TLR 8 (Fig 6C) and vaccine construct were analyzed by LigPlot software.

### 3.10. Simulation of the vaccine-receptor complex using molecular dynamics

With the aid of the iMOD server, we run a molecular dynamics simulation to assess the stability and physical motions of the vaccine-TLR3, 7, and 8 docked complex (López-Blanco, J. R.,

**Table 8. List of residue pairs in the vaccine construct that have the ability to create disulfide bonds.**

| Res1 Chain | Res1 Seq # | Res1 AA | Res2 Chain | Res2 Seq # | Res2 AA | Chi3 | Energy | Sum B-Factors |
|---|---|---|---|---|---|---|---|---|
| **A** | **335** | **GLY** | **A** | **373** | **GLY** | **-84.41** | **0.98** | **0** |
| **A** | **24** | **GLN** | **A** | **28** | **ASP** | **-81.06** | **1.05** | **0** |
| A | 560 | LEU | A | 566 | SER | 107.67 | 1.9 | 0 |
| A | 69 | PHE | A | 113 | THR | -92.63 | 2.26 | 0 |
| A | 413 | SER | A | 471 | PHE | 92.26 | 2.37 | 0 |
| A | 155 | PRO | A | 159 | PHE | 125.05 | 2.66 | 0 |
| A | 233 | SER | A | 286 | MET | -95.4 | 2.71 | 0 |
| A | 71 | VAL | A | 120 | ILE | 97.98 | 2.9 | 0 |
| A | 325 | ASP | A | 367 | ASP | 93.88 | 3.09 | 0 |
| A | 195 | TYR | A | 204 | THR | -111.79 | 3.14 | 0 |
| A | 74 | PRO | A | 81 | SER | 117.92 | 3.54 | 0 |
| A | 136 | ALA | A | 140 | ALA | 99.44 | 3.66 | 0 |
| A | 256 | PHE | A | 323 | SER | 104.89 | 3.87 | 0 |
| A | 560 | LEU | A | 565 | ASP | -97.12 | 4.02 | 0 |
| A | 300 | ASP | A | 341 | THR | 111.57 | 4.22 | 0 |
| A | 244 | ALA | A | 250 | TYR | -114.4 | 4.24 | 0 |
| A | 273 | GLY | A | 276 | PHE | 122.61 | 4.37 | 0 |
| A | 292 | PRO | A | 333 | GLY | 83.85 | 4.37 | 0 |
| A | 34 | HIS | A | 88 | ARG | -108.2 | 4.39 | 0 |
| A | 287 | ILE | A | 330 | PHE | -113.98 | 4.45 | 0 |
| A | 328 | ILE | A | 367 | ASP | 118.56 | 4.61 | 0 |
| A | 248 | ALA | A | 298 | TYR | 100.21 | 4.78 | 0 |
| A | 457 | PRO | A | 462 | HIS | 112.29 | 5.26 | 0 |
| A | 185 | ASN | A | 217 | GLU | 105.09 | 5.59 | 0 |
| A | 287 | ILE | A | 331 | GLY | 124.62 | 5.75 | 0 |
| A | 443 | ILE | A | 446 | SER | 95.94 | 5.75 | 0 |
| A | 248 | ALA | A | 301 | ALA | 69.85 | 6.03 | 0 |
| A | 244 | ALA | A | 247 | ALA | -63.08 | 6.17 | 0 |
| A | 420 | ILE | A | 465 | LYS | 97.08 | 6.2 | 0 |
| A | 306 | ARG | A | 308 | ASN | 117.24 | 6.33 | 0 |
| A | 312 | PRO | A | 316 | VAL | 87.43 | 6.59 | 0 |
| A | 489 | SER | A | 492 | GLY | 69.6 | 6.95 | 0 |

Aliaga, 2014). Figs 7–9 illustrate the main-chain deformability for TLR 3, TLR 7, and TLR 8, respectively. Normal mode analysis indicates regions with high deformability when hinges are present, and B-factor values are proportional to root mean square values. Low RMSD and highly correlated areas in heat maps demonstrated the improved interaction of each residue. These findings demonstrate the structure of protein with its MNA mobility, the low deformability present in every residue, the B-factor, the Eigen value, the variance in both green and red, the elastic network, and the covariance of the complex.

## 3.11. Simulated immune responses indicate an efficient response via antibodies and cytokines

The immunological response that was produced by the *in silico* vaccine construct was shown to be effective in the simulation results obtained from C-ImmSim. In order to mimic the body's immune response, we gave the vaccine to the animals twice (Fig 10). The subsequent responses were of a

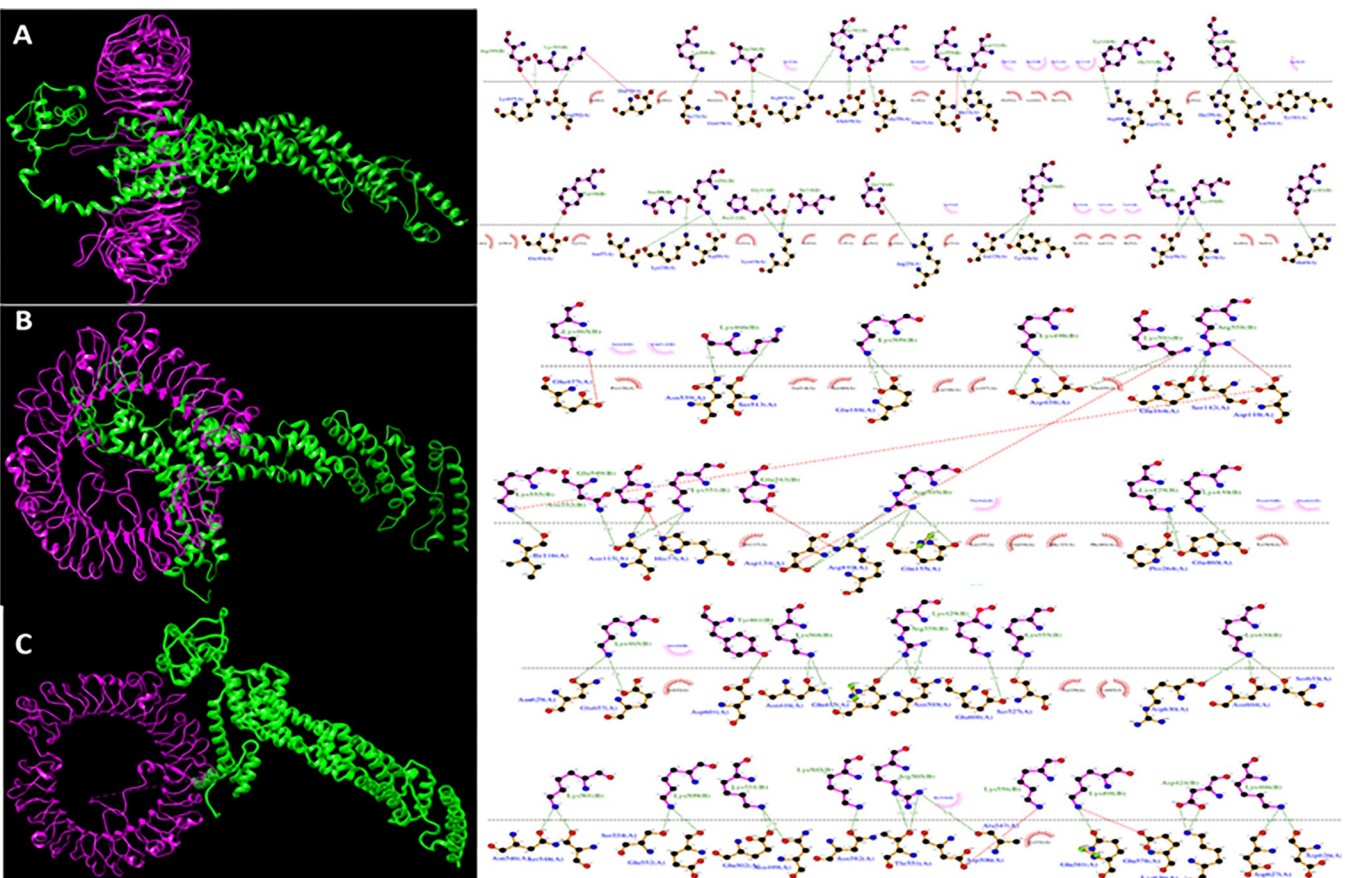

**Fig 6. The docked complex of the multi-epitope vaccine and TLR was visualized using the UCSF Chimera software and interaction in the docked complex has been shown using Ligplot software.** (a) docked complex and interaction for TLR3 with vaccine construct (b) docked complex and interaction for TLR7 with vaccine construct (c) docked complex and interaction for TLR8 with vaccine construct.

greater quality than the initial ones. The amount of immunoglobulin (IgM) that was created was greater than that of IgG. Following the antigen decrease, the levels of immunoglobulins rose to a high level. This rise could point to the formation of immunological memory as a result of exposure to the antigen, which would show that the individual had practised their immunity. Indicative of these findings was an increase in the quantity of IgM as well as expression of IgG1+IgG2, IgM, and IgG+ IgM. In addition, the upregulation of both cytotoxic T cells and Helper T cells as a result of the multi-epitope vaccine design provided further confirmation of an adequate immune response. Because of the presence of B-cell isotypes over an extended period of time, isotype switching and the creation of memories in the B cell population are both evidenced. In addition, there was a rise in the number of CTL and HTL cells associated with the formation of memories. In addition to this, macrophage activity increased, although dendritic cell activity remained unchanged. In addition to that, the levels of IFN- beta and TGF beta were both elevated.

### 3.12. *In silico* vaccine cloning in pcDNA™3.1/V5-His-TOPO⑭ expression vector

Jcat server codon optimized *in silico* vaccine candidate for *E.coli* protein expression. The vaccine has 1716 nucleotides and 65.09% GC. SnapGene program was use to clone cloned the design into pcDNA™3.1/V5-His-TOPO⑭ (Fig 11).

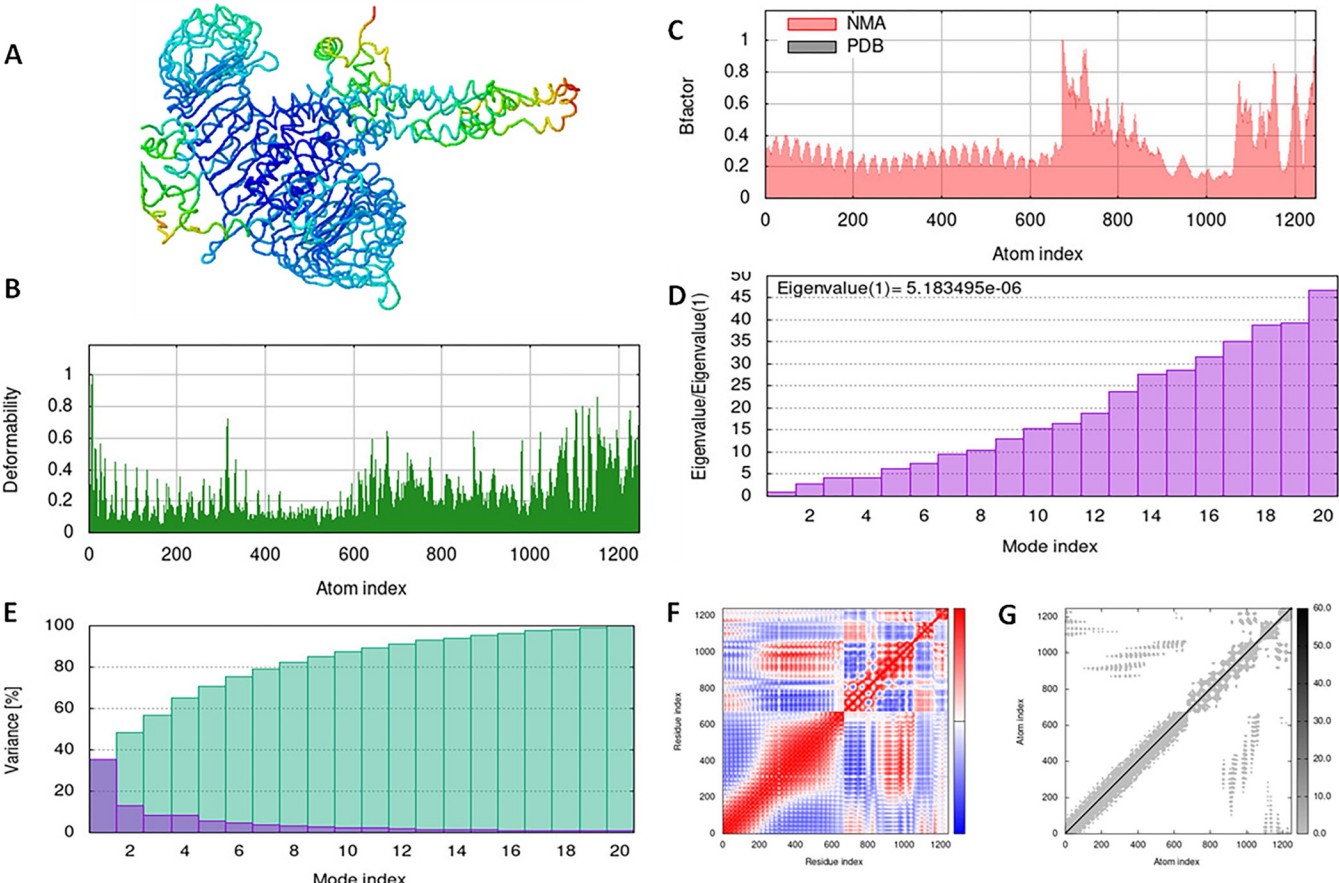

**Fig 7. Molecular dynamics simulation of vaccine construct and TLR3 complex by iMODS server.** (A) vaccine construct and TLR3 docking complex. (B) Main-chain deformability. (C) B-factor values. (D) The eigenvalue. (E) Variance. (F) Co-variance map. (G) Elastic network of model.

## 4. Discussion

After an increase in the global health emergence that has been created by various fatal viruses, it is necessary that progress be made in the quest for an acceptable treatment for the diseases. Vaccination is consistently regarded as one of the most successful approaches available for the prevention of infectious diseases caused by viruses [38]. The production of vaccinations is widely regarded as one of the most significant steps forward in efforts to cut morbidity and death rates around the world. Not only does it prevent the start of a variety of ailments, but it also identifies a doorway for the disease's removal while simultaneously reducing its toxicity [39]. Antigens that are exploited in vaccines do not necessarily need to be virulence factors [40]. Despite the fact that most virulence gene products are immunogenic and contribute to acquired immunity protection against disease. Antigens in vaccines can be any infectious disease-causing agent that is known to science. According to Ali et al., protein antigenicity refers to its capacity to activate an immune response to the organism that it is a part of. Antigenicity of a protein is a necessary condition before the protein can be utilized as a component of a vaccine [41].

The present study focused on the identification of the potential epitopes for the development of the effective vaccine using immunoinformatic methods {Bahrami, 2019 64 /id}. In the development of epitope vaccines, the precise identification of epitopes is a crucial stage, and

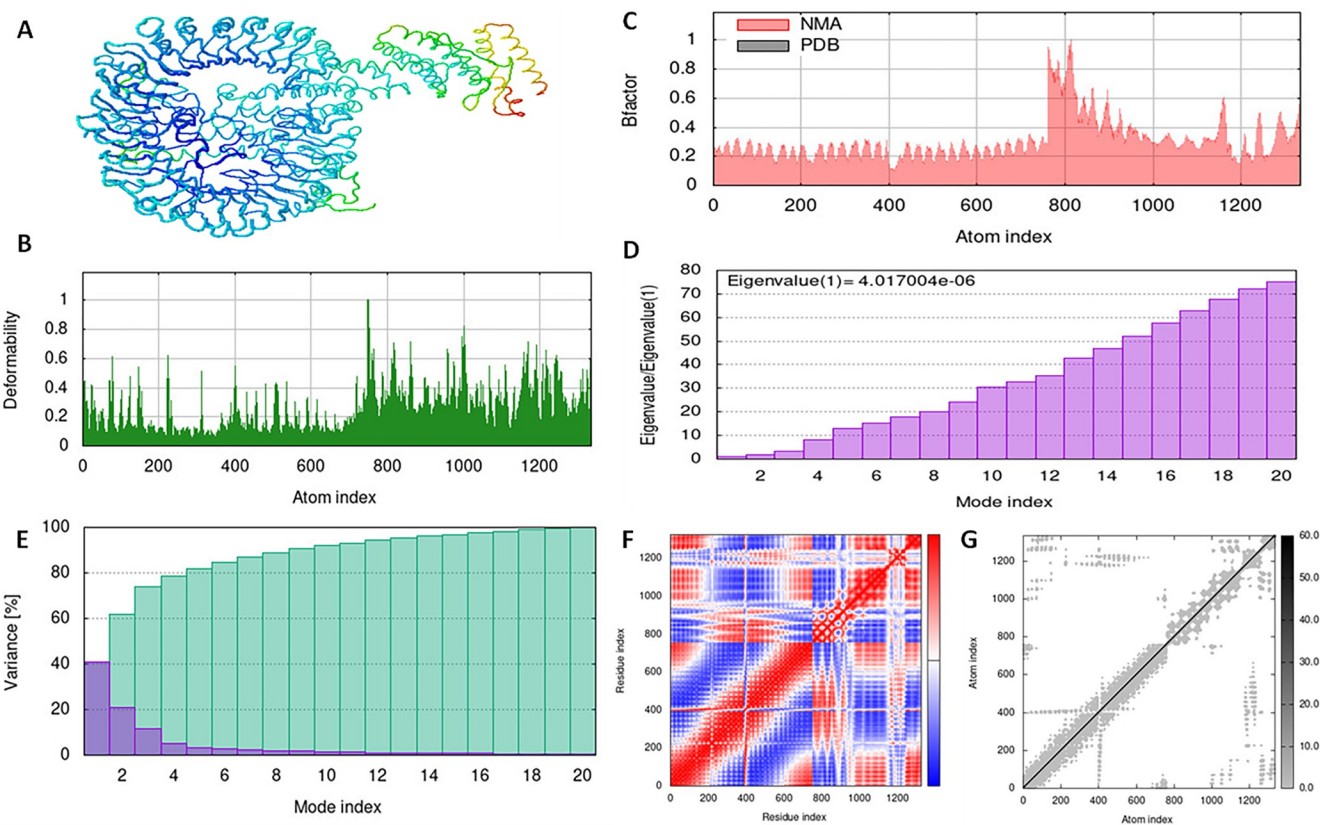

**Fig 8. Molecular dynamics simulation of vaccine construct and TLR7 complex by iMODS server.** (A) vaccine construct and TLR7 docking complex. (B) Main-chain deformability. (C) B-factor values. (D) The eigenvalue. (E) Variance. (F) Co-variance map. (G) Elastic network of model.

novel approaches to combining epitopes are currently under investigation [42]. According to research carried out by Galluzzi et al., the immune response is set off when antigens bind to the ligands or receptors that are located on the surfaces of immune cells [43]. It is generally agreed upon that the G and N protein is a necessary antigen for the production of the NiV vaccine. Several vaccines are currently being developed that use NiV G protein or G protein plus F protein as antigens [44–46]. These vaccines are currently in the process of being tested on humans. It is anticipated that these vaccines will be available in the not-too-distant future. According to the findings, G protein was the primary factor in the production of the antibodies that were capable of neutralizing the virus. The protein subunit platform serves as the basis for the Equivac HeV vaccine, which is the only Hendra virus vaccine that can be used clinically at this time and holds a valid license to do so. This proved that the proposed technology was a trustworthy and established platform, which could be utilized in the production of vaccines against either the Hendra or the NiV [20].

According to a previous study, T-cell epitopes are necessary for MHC molecules to work together and trigger the immune system. When predicting potent T-cell epitopes, it is imperative to select epitopes that are likely to be fixed by MHC [47,48]. In addition, determining which cells have CD4+ and CD8+ T receptors is essential for the advancement of MSV [49,50]. The incorporation of linkers into specialized sequences has the potential to enhance the construction of vaccines. In the past, many studies have shown that the addition of GPGPG and AAY linkers [41,51,52] between predicted HTL and CTL epitope sequences generated junctional immunogenicity, thereby enabling the logical design and construction of a

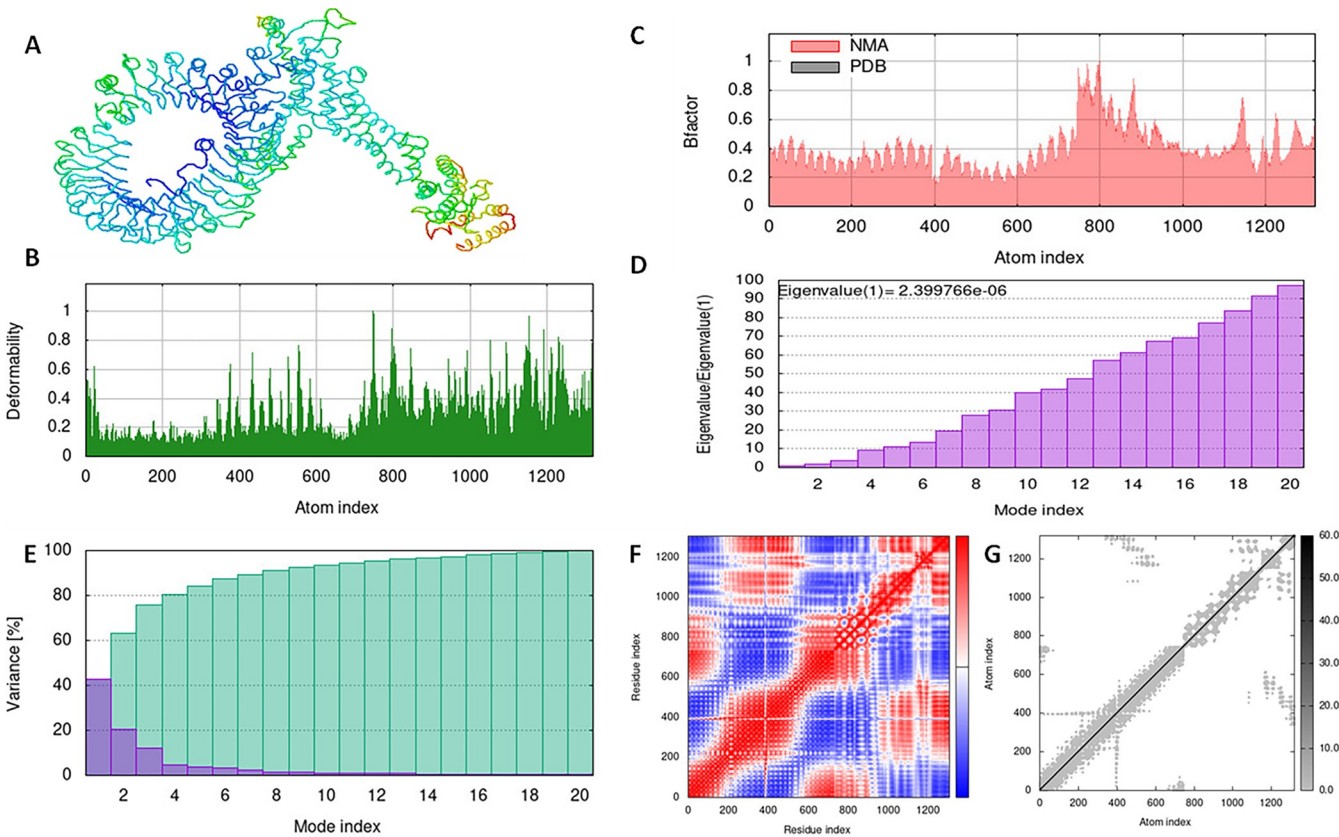

**Fig 9. Molecular dynamics simulation of vaccine construct and TLR8 complex by iMODS server.** (A) vaccine construct and TLR3 docking complex. (B) Main-chain deformability. (C) B-factor values. (D) The eigenvalue. (E) Variance. (F) Co-variance map. (G) Elastic network of model.

potent poly-epitope vaccine [53]. In their study, Arai et al. [54] utilized the EAAAK linker to connect epitopes and adjuvant, thereby enhancing the bioactivity of the fused protein, achieving a high level of expression, and bolstering the stability of the vaccine construct. Bazan et al. developed a multi-epitope subunit vaccine against the Ebola virus utilizing T-cells. Antigenic epitopes predicted using the Immune Epitope Database (IDEB) were incorporated into the development of a vaccine candidate that exhibited immunogenicity upon expression in rodents [55]. Antibodies are able to make a contribution to the eradication of viral loads and the neutralization of pathogen-associated antigens. The ABCpred service was used in order to analyze the antigenic proteome and add epitopes into the design of the vaccine that are capable of activating B cells. Various studies showed the Cholora toxin (CT) adjuvant enhanced the immune response to influenza vaccines in mice and provided cross-protection against heterologous influenza strains and *Helicobacter pylori* [53,56]. Overall, these studies demonstrate the potential of CT as an adjuvant for multi-epitope vaccines against various diseases. CT's ability to induce mucosal immunity and enhance immune responses makes it an attractive candidate for the development of novel vaccines.

The proposed vaccine construct had molecular weight was 63.27 kDa, and its theoretical pI was found to be 9.79, which means it is alkaline. The predicted half-life of our vaccine candidate (30 hours in human reticulocytes, more than 20 hours in yeast, and more than 10 hours in E.) was longer than that of the vaccine made by Waqas et al. [57], which was 7.2 hours in reticulocytes (in vitro) and similar to previous studies [52,58]. This means that the vaccine is

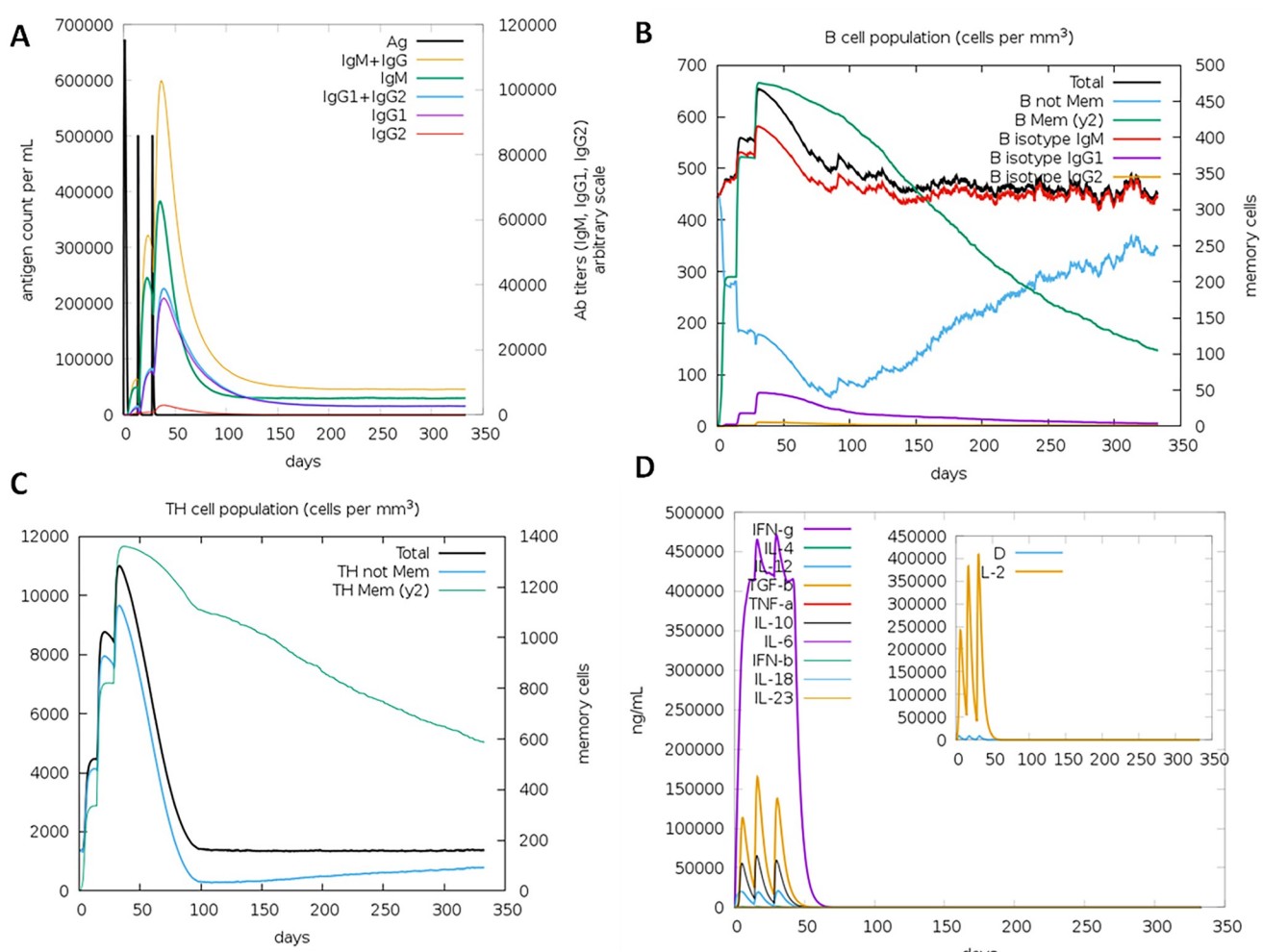

**Fig 10. In silico immune simulation of VC1 by IMMSIM server.** (A) Immunoglobulin production upon antigen exposure. (B) Population of B lymphocytes, y2 represents scale of memory B cells. (C) Population of T lymphocytes, y2 represents scale of memory T cells. (D) Concentration of cytokines and interleukins.

in contact with the immune system for a long time [59]. The vaccine instability index was 28.56, indicating its stability, which was good than some previous studies Kadam et al. 2020; Shafi Mahmud et al., 2021. In general, a protein with an instability index of less than 40 is considered stable [59]. The vaccine's aliphatic index was found to be 78.08, which means it stays stable at a lot of different temperatures [60]. The aliphatic index value of this vaccine was higher than the aliphatic index of the vaccine designed in works by Sukrit Srivastava, et al (48.03) [61], Waqas et al. (74.63) [57], and less than Zaib et al. (79.90) [62], and Shantier et al. (65.75) [63]. It was found that the multi-epitope vaccine has a GRAVY score of -0.218. This score shows how hydrophilic the vaccine is and is linked to how well proteins dissolve. A negative value means that the vaccine interacts better with water molecules [58].

Understanding the secondary and tertiary structures of proteins is necessary for the production of vaccines, as stated by Meza et al. [53]. According to what was discovered in the research, coils make about fifty percent of the secondary structure of the protein. The natural antigenicity of the unfolded native secondary proteins is reflected in the helices, which account for the vast majority of the coils and are composed of helices. The immune system is able to recognize the spiral shape once it has been folded back into its original state. According to the

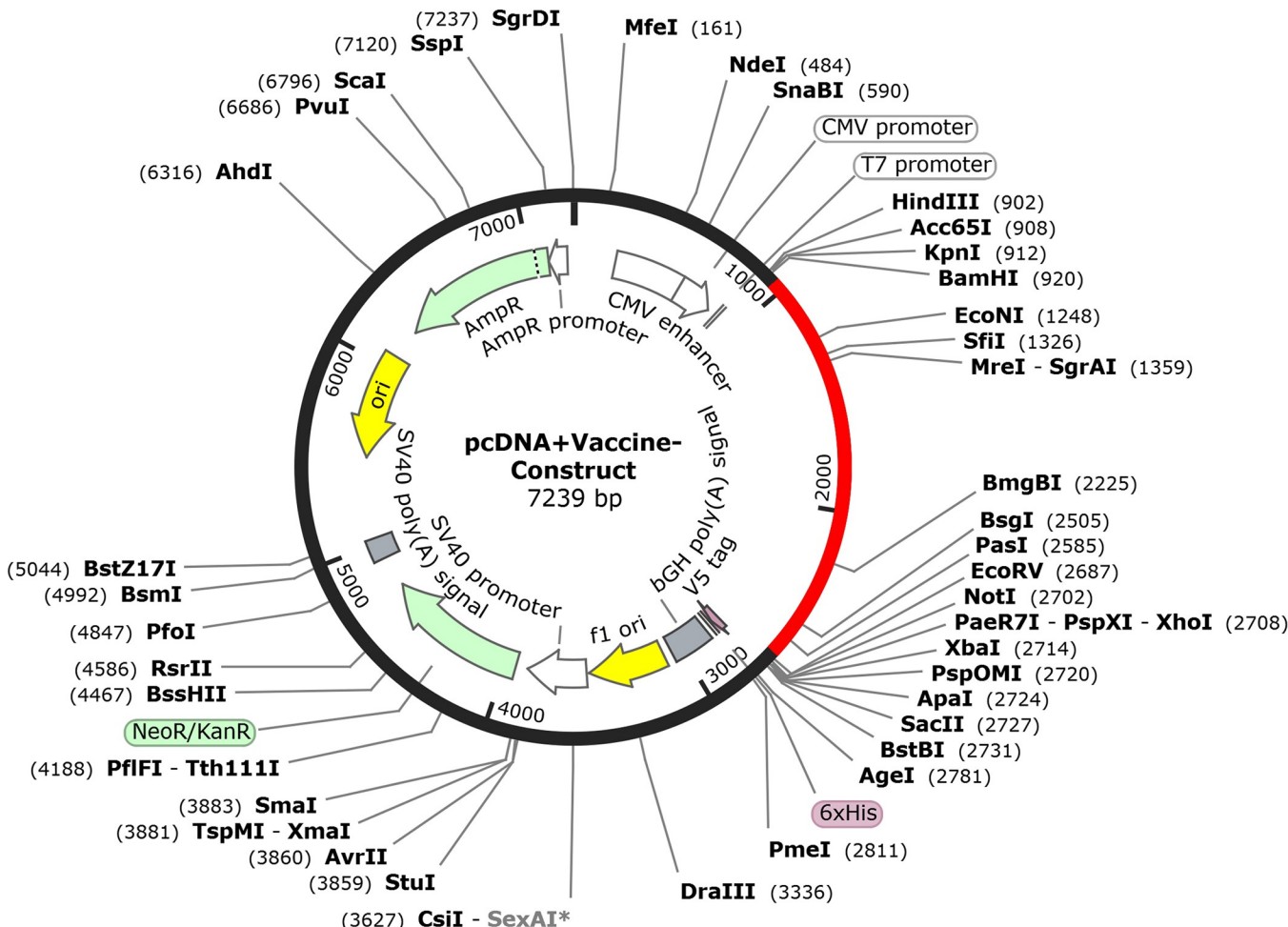

**Fig 11. In silico cloning of multiepitope subunit vaccine sequence into pcDNA™3.1/V5-His-TOPO⑧ expression vector by SnapGene tool, red color part represents vaccine sequence and black circle represent vector sequence.**

findings of the Ramachandran plot, the 3D model of the vaccine, after being changed, revealed elements that were acceptable and was significantly improved. This is evidenced by the fact that the model was significantly better. The Ramachandran plot reveals that 90.3% of residues begin in preferred domains, whereas only 8.3% of residues begin in allowed regions. This disparity indicates that the output of the model as a whole is satisfactory. The immuno-reactive need in a serological sample, is important for establishing whether or not a vaccine candidate is real.

TLRs are specifically accountable for the detection of viruses and the initiation of innate immunity, which is succeeded by adaptive immunity. Thus, they play a crucial role in innate immunity. TLR3, TLR7, and TLR8 are capable of identifying ds/ssRNA, a prevalent intermediary in the replication process of numerous RNA viruses [64,65]. Docking investigations play a pivotal role in comprehending the molecular interactions that regulate immune responses. By simulating the initial recognition stages of the immune system, signaling pathways are stimulated, which ultimately contributes to the development of a robust immune response. In order to enhance the immune response, vaccine adjuvants frequently target TLRs for stimulation. Gaining insight into the manner in which vaccine constructs engage with particular TLRs can enhance the effectiveness of vaccines, customize immune responses, and facilitate the

development of personalized vaccines. These studies additionally offer mechanistic under-standings regarding the mechanisms by which vaccine constructs stimulate TLRs, thereby improving immunogenicity and informing the development of constructs that stimulate more robust and targeted immune responses [64]. In addition, the MD simulation system was used to investigate the connection of TLR on immune cells via the vaccine-TLR docking complex. This investigation was carried out in order togather more information. According to the results of our study, TLRs displayed a strong affinity for the newly designed vaccine. According to Crozat and Beutler [66], the interaction between TLR and the vaccination revealed that it had the potential to generate both innate and adaptive inflammatory responses. LigPlot+ was used to illustrate the intramolecular interactions between the vaccine design and TLR3, 7 and 8. The effectiveness of the vaccine design to induce an immune response is explained by an immunological simulation study. One of the requirements for becoming an effective vaccina-tion candidate is the induction of memory B-cells and T-cells [67,68]. The number of memory B-cells and T-cells grew with each administration of the vaccine construct, and this population level persisted after the third injection. The ability of vaccine designs to create an antiviral state is also demonstrated by the raised levels of IFN-γ and IL2. In general, IFN-γ participates in antiviral replication and serves as the primary effector molecule in cell-mediated immunity. *Escherichia coli* is the best host to use if you want to produce a larger amount of recombinant vaccine protein. Java The positive content of vaccine constructions for high-level protein pro-duction in the E. coli host was confirmed by the codon adaptability tool, which showed that the codon adaptability index for vaccine candidates was > 0.95% and that the GC content was 65%.

The traditional method of developing vaccines is an expensive and drawn-out process that necessitates growing the pathogen in a lab [69]. Without cultivating the pathogen, reverse vac-cinology uses computational analysis and genomic data to generate vaccines. Predicting T and B cell epitopes, antigen processing, antigenicity, allergenicity, toxicity prediction, and TLR-peptide docking are crucial steps in designing virus vaccines [70].

## 5. Conclusion

Immune informatics tools were used in this study to develop a multi-epitope NiV vaccine. There have only been a few studies that have identified epitopes that target T-cells and B-cells using NiV's attachment glycoproteins (G) and nucleocapids (N). The present NiV-G and NiV-NP, MHC I and MHC II epitope vaccine formulations are highly immunogenic and anti-genic. The adjuvant was added to further increase the construct's efficacy after the epitopes were coupled using the appropriate linkers. The ability of the chimeric vaccine peptide to elicit an immune response must be tested *in vitro* and *in vivo*. It would facilitate the future creation of NiV vaccine that would benefit the intended population.

## Supporting information

**S1 File.**
(XLSX)

## Author Contributions

**Conceptualization:** Anoop Kumar.

**Writing – original draft:** Anoop Kumar.

**Writing – review & editing:** Gauri Misra, Sreelekshmy Mohandas, Pragya D. Yadav.

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
