## [Decision Letter · Decision Letter 0]

26 Jan 2024

PONE-D-23-41371Multi-Epitope Vaccine Design Using in silico analysis of Glycoprotein and Nucleocapsid of NIPAH virusPLOS ONE

Dear Dr. Kumar,

Thank you for submitting your manuscript to PLOS ONE. After careful consideration, we feel that it has merit but does not fully meet PLOS ONE’s publication criteria as it currently stands. Therefore, we invite you to submit a revised version of the manuscript that addresses the points raised during the review process.

We look forward to receiving your revised manuscript.

Kind regards,

Sheikh Arslan Sehgal, PhD

Academic Editor

PLOS ONE

5. Please include your tables as part of your main manuscript and remove the individual files. Please note that supplementary tables (should remain/ be uploaded) as separate "supporting information" files

Reviewers' comments:

Reviewer's Responses to Questions

**Comments to the Author**

1. Is the manuscript technically sound, and do the data support the conclusions?

Reviewer #1: Partly

Reviewer #2: Partly

2. Has the statistical analysis been performed appropriately and rigorously? 

Reviewer #1: N/A

Reviewer #2: N/A

3. Have the authors made all data underlying the findings in their manuscript fully available?

Reviewer #1: Yes

Reviewer #2: Yes

4. Is the manuscript presented in an intelligible fashion and written in standard English?

Reviewer #1: Yes

Reviewer #2: Yes

5. Review Comments to the Author

Reviewer #1: Dear authors please consider the mentioned comments related to manuscript entitled “Multi-Epitope Vaccine Design Using in silico analysis of Glycoprotein and Nucleocapsid of NIPAH virus”

1. Please mention in the manuscript the threshold, accuracy and URL for all used servers in epitopes selection and vaccine design. Also the references for all used servers should be mentioned in manuscript.

2. The conformational B cell epitopes should be selected.

3. This is a bioinformatics study, so it is better to change the section of materials and methods to methods.

4. Please, the authors answer to this question, what is the reason for doing docking with different TLRs (3, 7, and 8)?

5. It is suggested to refer to the following paper in the use section of the EAAAK, GPGPG and kk linkers.

https://pubmed.ncbi.nlm.nih.gov/37146081/

6. The references should be updated.

Reviewer #2: Overall, these suggestions aim to improve the clarity, accessibility, and impact of the manuscript, and I look forward to seeing these enhancements in the revised version. Thank you for your valuable contribution to the field.

1. Authors should use more specialized words for the manuscript. Unfortunately, inappropriate terms such as "projected epitopes" are used in some parts of the manuscript. The manuscript must be checked and corrected by an expert.

2. The words “in silico”, “in vivo” and “in vitro” should be written in italics throughout the manuscript.

3. Figure 1 is not very satisfactory, it is suggested to draw an attractive flowchart to present the working method.

4. In lines 104, 105, 172 , 384, and Table 4 after IFN, put gamma characters.

5. In the materials and methods section, the authors must provide sources for all used servers and give them citations.

6. For some sentences like " As an adjuvant, CTB has been found to increase the production of antigen-specific antibodies and stimulate the activation of T-cells, leading to a more robust and effective immune respons", no reference is mentioned. Please be sure to mention the relevant source for all these sentences.

7. Among the physicochemical properties, the instability index parameter is an important and decisive parameter that the authors did not report.

8. In order to complete the article, the authors must predict discontinuous B-cell epitope on the three-dimensional structure of the vaccine and present the results in the manuscript.

9. To complete the paper, the authors must perform disulfide engineering of the 3D structure of the vaccine and present the results in the manuscript.

10. For the richness of the study, the authors should compare their results with the results of other studies in the field of vaccine design, it is suggested to use the following studies for this purpose.

https://doi.org/10.1371/journal.pone.0286224

https://doi.org/10.1080/07391102.2023.2258403

https://doi.org/10.3390%2Fvaccines11020263

https://doi.org/10.2174/1573409919666230612125440

https://doi.org/10.1007/s12033-023-00949-y

https://doi.org/10.1038/s41598-022-11851-z

6. PLOS authors have the option to publish the peer review history of their article (what does this mean?). If published, this will include your full peer review and any attached files.

Reviewer #1: **Yes: **Shirin Mahmoodi

Reviewer #2: No

---

## [Author Response · Author response to Decision Letter 0]

24 Feb 2024

Response to the Reviewers

The authors are grateful to the Reviewers for the time devoted to the review of the manuscript entitled “Multi-Epitope Vaccine Design Using in silico analysis of Glycoprotein and Nucleocapsid of NIPAH virus” (ID: PONE-D-23-41371). We have carefully considered the comments and observations as well as the suggestions given in the reports and suitably modified the original text in order to address them to our best. The suggested changes are highlighted in track change mode in the revised manuscript. 

Detailed responses to the different points raised by the Reviewers are given below.

1. Please mention in the manuscript the threshold, accuracy and URL for all used servers in epitopes selection and vaccine design. Also the references for all used servers should be mentioned in manuscript.

Ans: As suggested by the reviewer, the needful has been done at the required places. 

2. The conformational B cell epitopes should be selected.

Ans: As suggested by the reviewer, conformational B cell epitopes has been predicted using ellipro tool (http://tools.iedb.org/ellipro/) of IEDB and results has been incorporated in the manuscript.

3. This is a bioinformatics study, so it is better to change the section of materials and methods to methods.

Ans: As suggested by the reviewer, the materials and methods now changed as Methodology. 

4. Please, the authors answer to this question, what is the reason for doing docking with different TLRs (3, 7, and 8)?

Ans: TLRs are specifically accountable for the detection of viruses and the initiation of innate immunity, which is succeeded by adaptive immunity. Thus, they play a crucial role in innate immunity. TLR3, TLR7, and TLR8 are capable of identifying ds/ssRNA, a prevalent intermediary in the replication process of numerous RNA viruses (Chuang, T. H., and R. J. Ulevitch. 2000.; Lester SN, 2014). Docking investigations play a pivotal role in comprehending the molecular interactions that regulate immune responses. By simulating the initial recognition stages of the immune system, signaling pathways are stimulated, which ultimately contributes to the development of a robust immune response. In order to enhance the immune response, vaccine adjuvants frequently target TLRs for stimulation. Gaining insight into the manner in which vaccine constructs engage with particular TLRs can enhance the effectiveness of vaccines, customize immune responses, and facilitate the development of personalized vaccines. These studies additionally offer mechanistic understandings regarding the mechanisms by which vaccine constructs stimulate TLRs, thereby improving immunogenicity and informing the development of constructs that stimulate more robust and targeted immune responses (Lester SN, 2014). 

5. It is suggested to refer to the following paper in the use section of the EAAAK, GPGPG and KK linkers. https://pubmed.ncbi.nlm.nih.gov/37146081/

Ans: As suggested by the reviewer, needful has been done.

6. The references should be updated.

Ans: As suggested by the reviewer, references have been updated. 

Reviewer #2: Overall, these suggestions aim to improve the clarity, accessibility, and impact of the manuscript, and I look forward to seeing these enhancements in the revised version. Thank you for your valuable contribution to the field. 

1. Authors should use more specialized words for the manuscript. Unfortunately, inappropriate terms such as "projected epitopes" are used in some parts of the manuscript. The manuscript must be checked and corrected by an expert.

Ans: Thanks for the reviewer comment, the terms have been changed and needful has been done.

2. The words “in silico”, “in vivo” and “in vitro” should be written in italics throughout the manuscript.

Ans: Thanks for the reviewer comment, “in silico”, “in vivo” and “in vitro” has been change to italics. 

3. Figure 1 is not very satisfactory; it is suggested to draw an attractive flowchart to present the working method.

Ans: Thanks for the reviewer comment, figure has been changed and flowchart has been incorporated.

4. In lines 104, 105, 172, 384, and Table 4 after IFN, put gamma characters.

Ans: Thanks for the reviewer comment, the gamma character has been incorporated as required places. 

5. In the materials and methods section, the authors must provide sources for all used servers and give them citations.

Ans: Thanks for the reviewer comment, sources for all servers used in the study and their citations has incorporated in methodology section. 

6. For some sentences like " As an adjuvant, CTB has been found to increase the production of antigen-specific antibodies and stimulate the activation of T-cells, leading to a more robust and effective immune response", no reference is mentioned. Please be sure to mention the relevant source for all these sentences.

Ans: Thanks for the reviewer comment, the citations have been incorporated at the required places in the manuscript.

7. Among the physicochemical properties, the instability index parameter is an important and decisive parameter that the authors did not report.

Ans: Thanks for the reviewer comments, instability index parameter has been incorporated in line no 260 “computed instability index of 28.56 indicates that the construct is stable”

8. In order to complete the article, the authors must predict discontinuous B-cell epitope on the three-dimensional structure of the vaccine and present the results in the manuscript.

Ans: As suggested by the reviewer, conformational B cell epitopes has been predicted using ellipro tool (http://tools.iedb.org/ellipro/) of IEDB and results has been incorporated in the manuscript.

9. To complete the paper, the authors must perform disulfide engineering of the 3D structure of the vaccine and present the results in the manuscript.

Ans: Thanks for the reviewer comments, disulfide engineering of the 3D structure of the vaccine has been predicted and result has been incorporated. “The vaccine construct's model identify 32 residue pairs that may potentially create a disulfide bond, as predicted by the Disulfide by Design 2.13 server (Table 8). As per study by Craig DB, et al., for the formation of disulfide bonds the χ3 angle lies between -87˚ and +97˚ and energy value less than 2.2 kcal/mol, on the basis of the this only two amino acid pair (335-GLY-373-GLY & 24-GLN-28-ASP) were form the disulfide bond in the designed construct (Table 8). 

10. For the richness of the study, the authors should compare their results with the results of other studies in the field of vaccine design, it is suggested to use the following studies for this purpose.

Ans: Thanks for the reviewer valuable comments, new refences has been incorporated and the discussion has been modified.

---

## [Editor Report · Decision Letter 1]

29 Feb 2024

Multi-Epitope Vaccine Design Using in silico analysis of Glycoprotein and Nucleocapsid of NIPAH virus

PONE-D-23-41371R1

Dear Dr. Kumar,

We’re pleased to inform you that your manuscript has been judged scientifically suitable for publication and will be formally accepted for publication once it meets all outstanding technical requirements.

Kind regards,

Sheikh Arslan Sehgal, PhD

Academic Editor

PLOS ONE
---

## [Editor Report · Acceptance letter]

29 Apr 2024

PONE-D-23-41371R1 

PLOS ONE

Dear Dr. Kumar, 

I'm pleased to inform you that your manuscript has been deemed suitable for publication in PLOS ONE. Congratulations! Your manuscript is now being handed over to our production team.

Kind regards, 

on behalf of

Dr Sheikh Arslan Sehgal 

Academic Editor

PLOS ONE